

# Differences in aerosol and cloud properties along the central California coast when winds change from northerly to southerly

Kira Zeider[1], Grace Betito[2], Anthony Bucholtz[3], Peng Xian[4], Annette Walker[4], Armin Sorooshian[1,2*]

[1]Department of Chemical and Environmental Engineering, University of Arizona, Tucson, Arizona, 85721, USA
[2]Department of Hydrology and Atmospheric Sciences, University of Arizona, Tucson, Arizona, 85721, USA
[3]Department of Meteorology, Naval Postgraduate School, Monterey, California, 93943, USA
[4]Marine Meteorology Division, Naval Research Laboratory, Monterey, California, 93943, USA

*Correspondence to: Armin Sorooshian (armin@arizona.edu)



**Abstract.** Wind reversals resulting in southerly flow along the California coast are not well understood in terms of how aerosol and cloud characteristics change. This gap is addressed using airborne field measurements enhanced with data from space-borne remote sensing (Moderate Resolution Imaging Spectroradiometer), surface stations (Interagency Monitoring of Protected Visual Environments), and models (Navy Aerosol Analysis and Prediction System and Coupled Ocean/Atmosphere Mesoscale Prediction System), with a focus on sub- and supermicron aerosol, and cloud microphysical variables: cloud droplet number concentration ($N_d$), cloud optical thickness (COT), and cloud droplet effective radius ($r_e$). Southerly flow coincided with higher values of submicron aerosol concentration ($N_a$) and mass concentrations of species representative of fine aerosol pollution ($NO_3^-$ and nss-$SO_4^{2-}$) and shipping/continental emissions (V, oxalate, $NH_4^+$, Ni, OC, and EC). Supermicron $N_a$ did not change, however, heightened levels of acidic species in southerly flow coincided with reduced $Cl^-$:$Na^+$ suggestive of $Cl^-$ depletion in salt particles. Clouds responded correspondingly in southerly flow, with more acidic cloud water, higher levels of similar species as in the aerosol phase (e.g., $NO_3^-$, nss-$SO_4^{2-}$, $NH_4^+$, V), along with elevated values of $N_d$ and COT and reduced $r_e$ during campaigns with similar cloud liquid water paths. Case study flights help to visualize offshore pollution gradients and highlight the sensitivity of the results to the presence of widespread smoke coverage including how associated plumes have enhanced supermicron $N_a$. These results have implications for aerosol-cloud interactions during wind reversals, and have relevance for weather, public welfare, and aviation.



## 1    Introduction

The northeastern Pacific Ocean is one of the most heavily studied regions as it relates to aerosol-cloud interactions due to the persistent and spatially broad stratocumulus cloud deck that is influenced by a variety of emissions sources, notably shipping (Wood, 2012; Russell et al., 2013). One aspect of that region that warrants more attention is the predominant direction of lower tropospheric winds, as recent work has suggested that it can have significant implications for aerosol and cloud properties (Juliano et al., 2019a; 2019b). The wind direction along the North American west coast is influenced by its topography, namely the coastal mountains (e.g., National Research Council, 1992), and during the California (CA) warm season (April through September), it is primarily from the north along the coast. An important weather phenomenon during that season is the infrequent and short-lived (from one to several days) transition from northerly to southerly flow near the coast up to 100 km offshore (e.g., Nuss et al., 2000). Particularly, the northerly winds weaken (e.g., Winant et al., 1987; Melton et al., 2009) and eventually reverse. Along with a decrease in temperature and increases in pressure and cloud fraction (e.g., increases in low clouds and fog), there is also a change in overall wind speed: most northerlies (~75%) have a wind speed component less than 5 m s$^{-1}$ (Bond et al., 1996), whereas southerly "surges" are characterized by sudden increases in wind speed to 15 m s$^{-1}$ or greater (Mass and Albright, 1987). This is not a phenomenon that is unique to the U.S.; a handful of studies have noted these events along the coasts of South America (e.g., Garreaud et al., 2002; Garreaud and Rutllant, 2003), southern Africa (e.g., Reason and Jury, 1990), and even Australia (e.g., Holland and Leslie, 1986; Reason et al., 1999; Reid and Leslie, 1999).

These wind reversals – referred to as either coastally trapped disturbances (CTDs), coastally trapped wind reversals (CTWRs), stratus surges, or southerly surges, to name a few – have been studied since the 1970s (Gill, 1977; Dorman, 1985). There have been a fair number of publications discussing the dynamics and forcing mechanisms for such events (thoroughly reviewed by Nuss et al., 2000) primarily using data from buoys, radars, and research aircraft. Buoy (e.g., Bond et al., 1996) and satellite studies (e.g., Parish, 2000; Rahn and Parish, 2010) mainly discussed the topics related to mesoscale structure, while the research aircraft studies (e.g., Ralph et al., 1998; Rahn and Parish, 2007) have attempted to document physical characteristics of the wind reversal. For example, Rahn and Parish (2007) used sawtooth maneuvers to depict the vertical structure of the 22-25 June 2006 reversal through examining surface pressure, temperature, wind direction, wind speed, along-shore wind, and cross-shore wind. Additionally, there have been multiple studies attempting to model these wind reversals (e.g., Rogerson and Samelson, 1995; Guan et al., 1998; Skamarock et al., 1999; Mass and Steenburgh, 2000; Thompson et al., 2005) to better understand their initiation, propagation, and cessation. These studies found that CTDs are initiated by changes in synoptic-scale flow, particularly offshore, and that the coastal mountains dampen the flow, deepen the marine layer, and propagate a mesoscale coastal ridge of higher pressure northward that ultimately leads to the development of a coastally trapped southerly wind component.

However, there have been limited attempts to look into aerosol and cloud characteristics during a southerly surge (e.g., Juliano et al., 2019a; 2019b), and among them were studies that happened to encounter them by chance without these surges having been the study's focus (Crosbie et al., 2016; Dadashazar et al., 2020). Juliano et al. (2019a) was, to our best knowledge, the first study to focus on CTD aerosol-cloud interactions using 23 cases identified between 2004 and 2016 with buoy data and satellite imagery. They found notable differing characteristics between non-CTD (northerly flow) and CTD (southerly flow) conditions, with higher cloud droplet number concentration ($N_d$) and lower droplet effective radius ($r_e$) for CTD cases. Compared to non-CTD events, CTD events had $r_e$ values that were ~20-40% lower (i.e., differences often exceeding ~3 μm) and $N_d$ values (~250 cm$^{-3}$) that were almost twice as large in many areas. They attributed this to some combination of (i) mixing of sea salt particles into the boundary layer due to an observed wind stress-sea surface temperature cycle; (ii) offshore flow transporting continental aerosol into areas offshore of CA; and (iii) extended periods of time that southerly air spends in shipping lanes. Some continental sources they noted include agricultural emissions from the CA Central Valley, biogenic emissions from various major sources such as forests around Oregon and northern CA, smoke from biomass burning, and urban emissions from major CA cities such as Los Angeles, San Jose, Sacramento, and San Francisco. These sources have been confirmed in various studies conducted in coastal areas of central CA (Wang et al., 2014; Maudlin et al., 2015; Braun et al., 2017; Dadashazar et al., 2019; Ma et al., 2019). A subsequent study (Juliano et al., 2019b) analyzed three CTD events using satellite and aircraft observations, as well as numerical simulations. That study's usage of aircraft data was limited to cloud water composition, to support results from their previous study that non-CTD days were primarily influenced by marine sources like sea salt, whereas CTD days exhibited more relative influence from continental and shipping



(i.e., higher $SO_4^{2-}$ and $NO_3^-$) sources. Those studies noted that additional observations, specifically of an in situ nature,
were needed to confirm results that were mostly based on modeling and remote sensing.
The goal of this study is to contrast aerosol and cloud characteristics between southerly and northerly flow
regimes in the lower troposphere (below 3 km) offshore of central CA. Note that we do not focus here on
meteorological and large-scale features associated with wind reversals and do not classify events based on whether
they are CTDs but focus exclusively on boundary layer wind direction. As a way to address the shortage of in situ
observational data used for this research application, an important inventory of airborne data are leveraged that have
been collected over the last two decades (Sorooshian et al., 2018) that afford increased statistics of southerly flow
cases relative to Juliano et al. (2019b). Such cases are difficult to sample owing to their lower frequencies (Table 1)
compared to days with northerly flow and because aircraft flights do not occur each day, so some southerly cases are
missed during airborne campaigns. In total, 17 days of data exist from Naval Postgraduate School (NPS) Twin Otter
campaigns coinciding with southerly flow, with some days including multiple flights. One thing that has yet to happen
in past studies is to use in situ data to compare more than just cloud water composition but also relevant variables such
as aerosol number concentration ($N_a$) and $N_d$, which is crucial to intercompare with satellite data and put previous
speculations about aerosol and cloud responses to southerly flow on sturdier ground. As the aircraft data are still
limited, we complement the analysis with other datasets, including those from satellite remote sensors, models, and
surface stations.
The structure of this paper is as follows: Sect. 2 reports on methods used; Sect. 3 shows results beginning
with a discussion of how well a model can represent southerly winds, followed by assessing how well the datasets
show more fine pollution during southerly days and if clouds respond accordingly with the usual chain of events
associated with the Twomey effect (Twomey, 1974) whereby clouds have more but smaller drops at similar liquid
water path; and Sect. 4 provides conclusions. The results of this work have implications for numerous societal and
environmental factors sensitive to aerosol and cloud characteristics such as transportation (especially aviation),
agriculture, biogeochemical cycling of nutrients and contaminants, and coastal ecology (Dadashazar et al., 2020).
**2        Methods**
This study relies on the use of multiple datasets to examine how aerosol and cloud characteristics vary
between traditional northerly flow along the CA coastline as compared to less common southerly flow periods. This
study was initially inspired by airborne field measurements (Table 1) whereby on a few opportune flight days,
southerly flow was encountered off the CA coast. Because these events were rare in comparison to the majority of
flights with northerly flow (Southerly Winds % in Table 1), several campaigns worth of data are compiled to build
more statistics of southerly flow days. The airborne data used here are all from summer periods, which is when most
field studies have focused on this region to investigate aerosol-cloud interactions (e.g., Russell et al., 2013) allowing
for easier intercomparison for interested readers. We enhance statistics by also conducting complementary analyses
with data obtained from space-borne remote sensing, surface-based stations, and models. Below we first describe the
airborne datasets, followed by the wind classification method, and then descriptions of the models, surface data, and
satellite data.
**2.1      Airborne Field Missions**
This study utilizes data from six airborne missions based out of Marina, CA (Fig. 1) using the Naval
Postgraduate School (NPS) Twin Otter aircraft. The scientific target of these campaigns included a mix of aerosol-
cloud interactions, aerosol microphysical processes, and characterization of wildfire emissions: the Eastern Pacific
Emitted Aerosol Cloud Experiment (E-PEACE), the Nucleation in California Experiment (NiCE), the Biological and
Oceanic Atmospheric Study (BOAS), the Fog and Stratocumulus Evolution Experiment (FASE), the Marine Aerosol
Cloud And Wildfire Study (MACAWS), and the California Smoke Mission (CSM) (Table 1). Another Twin Otter
mission from 2019 (Monterey Aerosol Research Campaign - MONARC) is not included in this analysis due to the
lack of southerly flow days sampled during the campaign. The research flight (RF) paths for each campaign are shown
in Fig. 1. In some instances, multiple flights were conducted on a single day, either to capture time-sensitive
atmospheric features or to collect data beyond the endurance limit of the instrumented aircraft. For those days, RFs
are assigned the same number but are distinguished with endings 'A,' 'B,' and 'C,' for successive flights, respectively.
E-PEACE and NiCE had the most cases of southerly flow owing partly to those campaigns having had the most flights:
five out of 30 flights for E-PEACE; four out of 23 flights for NiCE. BOAS also had four flights with southerly flow



(out of 15 flights) but they were spread across two flights days as compared to E-PEACE and NiCE whose southerly
flights were all on distinct days.
The Twin Otter flew at ~55 m s$^{-1}$ and conducted measurements during level legs and sounding profiles, over
both the land and the ocean, and within and above the boundary layer during flight periods ranging from one to five
hours. Additional information regarding aircraft and flight characteristics, as well as the general flight strategy is
summarized in Sorooshian et al. (2019). The general area of focus in this study was within the following range of
coordinates, with many of the results specifically targeting just the ocean areas in this spatial domain: 35.31° N –
40.99° N, 125.93° W – 118.98° W.
**Table 1: Summary of NPS Twin Otter campaigns used in this study, including dates, number of RFs per campaign, RFs**
**that are categorized as having had southerly flow, and percentage of southerly days during the campaign period (including**
**all days in those months and not just RF days). Days are categorized as having southerly flow based on the analysis in Sect.**
**2.2.**

| Campaign | Dates | Total RFs | RF # (Flight Date) with Southerly Winds | Southerly Winds % (# Southerly days / Total days in period ) |
|---|---|---|---|---|
| E-PEACE | 07/08 – 08/18/2011 | 30 | RF11 (07/23), RF12 (07/24), RF14 (07/27), RF15 (07/28), RF16 (07/29) | 12.90% (8/62) |
| NiCE | 07/08 – 08/07/2013 | 23 | RF7 (07/16), RF8 (07/17), RF9 (07/18), RF16 (07/29) | 14.52% (9/62) |
| BOAS | 07/02 – 07/24/2015 | 15 | RF10A & 10B (07/16), RF11A & 11B (07/17) | 32.26% (10/31) |
| FASE | 07/18 – 08/12/2016 | 16 | RF6A, 6B, & 6C (07/29) | 14.52% (9/62) |
| MACAWS | 06/21 – 07/12/2018 | 16 | RF12 (07/05), RF16 (07/12) | 4.92% (3/61) |
| CSM | 09/01 – 09/25/2020 | 14 | RF1 (09/01), RF5 (09/09), RF6 (09/10) | 13.33% (4/30) |


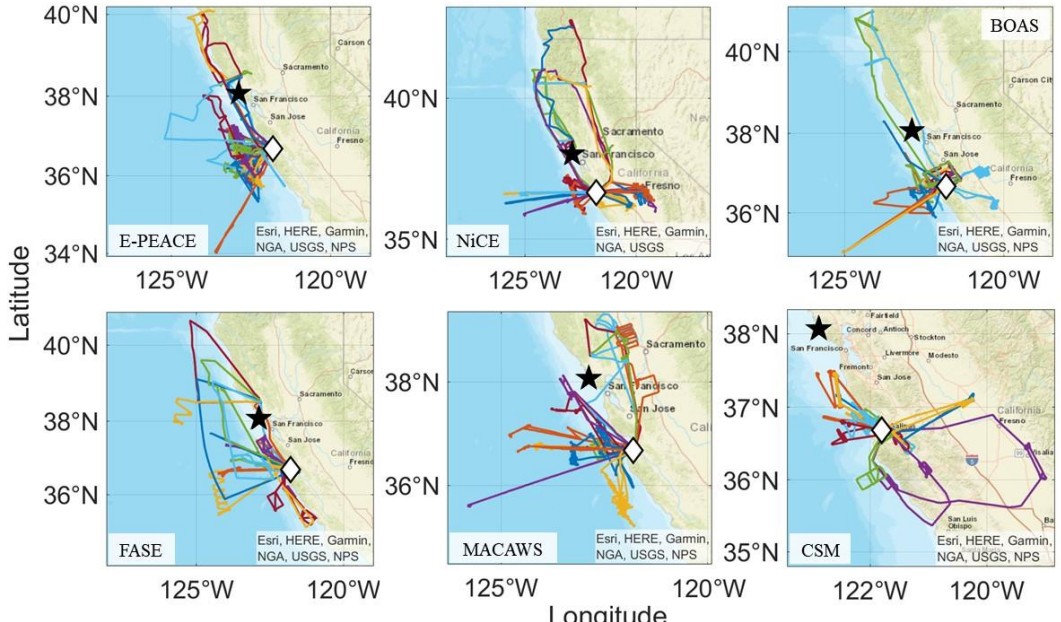

**Figure 1: Research flight paths for the six Twin Otter campaigns used in this study. The aircraft base at Marina, CA is**
**denoted by a white diamond, and the IMPROVE station used in this study is indicated by a black star (Pt. Reyes National**
**Seashore).**
**2.1.1   Twin Otter Instrumentation**
Table 2 summarizes the relevant instruments used for each Twin Otter mission pertinent to this work. More
extensive details about the instruments, and those not listed below such as relevant navigational and meteorological
instruments, are described in Sorooshian et al. (2018).
**Table 2: Summary of Twin Otter payload during the field campaigns used for this study. The six farthest right columns**
**show instrument availability for each campaign.**



| Instrument | Measured variable | Size range | Time resolution | E-PEACE | NiCE | BOAS | FASE | MACAWS | CSM |
|---|---|---|---|---|---|---|---|---|---|
| TSI Ultra-fine Condensation Particle Counter (CPC) 3025 | $N_{a>3nm}$ | > 0.003 μm | 1 s | X | X | X | X | X | X |
| TSI Condensation Particle Counter (CPC) 3010 | $N_{a>10nm}$ | > 0.01 μm | 1 s | X | X | X | X | X | X |
| PMS/DMT Passive Cavity Aerosol Spectrometer Probe (PCASP) | $N_{a0.1-1\mu m}$, $N_{a>1\mu m}$ | ~0.1 – 3.4 μm | 1 s | X | X | X | X | X | X |
| DMT Cloud and Aerosol Spectrometer - Forward Scattering (CASF) | $N_d$ | ~0.6 - 60 μm | 1 s | X | X | | X | X | X |
| PMS/DMT Forward Scattering Spectrometer Probe (FSSP) | $N_d$ | 1 - 46 μm | 1 s | | X | X | X | X | |
| ARI Aerosol Mass Spectrometer (AMS) | Speciated mass conc. | ~60 - 600 nm | < 15 s | X | X | X | | | |
| Mohnen Cloud Water Collector - pH, IC, ICPMS | pH, air-equivalent mass conc. | N/A | ~ 5 - 60 min | X | X | X | X | X | |

Condensation particle counters (CPCs; TSI, Inc.) were used to measure particle number concentrations for
diameters greater than 3 ($N_{a>3nm}$ or $N_{a3}$) and 10 nm ($N_{a>10nm}$ or $N_{a10}$), respectively, as well as the Passive Cavity Aerosol
Spectrometer Probe (PCASP; Particle Measuring Systems (PMS), Inc., modified by Droplet Measurement
Technologies (DMT), Inc.) for diameters between ~100 nm and 3.4 μm. The Cloud and Aerosol Spectrometer –
Forward Scattering (CASF; DMT, Inc.) measured the size distribution of larger particles and droplets between 0.6 –
60 μm for all missions except for BOAS when the Forward Scattering Spectrometer Probe (FSSP; PMS, Inc. modified
by DMT, Inc.) was used in its place. The cloud probes were calibrated before each field campaign to ensure
consistency between the instruments (Sorooshian et al., 2018). The CASF and FSSP size distributions were integrated
to determine total $N_d$ and liquid water content (LWC) when the aircraft was in cloud using the criterion of LWC
needing to exceed 0.02 g m$^{-3}$; all instances of LWC less than 0.02 g m$^{-3}$ were considered cloud-free and only considered
for quantification of aerosol variables such as total $N_a$ in different size ranges (Fig. S1). Additionally, RFs categorized
as southerly flow were filtered to only include data during periods when the horizontal wind direction was between
135° and 225°. A variety of statistics were calculated for the reported and derived variables (e.g., $N_{a>3nm}$, $N_{a>10nm}$, $N_{a10-}$
$_{100nm}$ ($N_{a>10nm}$ - $N_{a0.1-1\mu m}$), $N_{a0.1-1\mu m}$, $N_{a>1\mu m}$, the ratio of $N_{a3}$ to $N_{a10}$ ($N_{a3}$:$N_{a10}$), $N_d$, horizontal wind speed and direction)
in categories of interest including medians and minimum/maximum values. The mode was calculated for wind
direction for each RF as well as each overall campaign, since that statistic is assumed here to be a better representation
of typical wind directions rather than the median.
An Aerosol Mass Spectrometer (AMS; Aerodyne Research Inc. (ARI)) was used during some campaigns to
measure sub-micrometer (submicron) aerosol composition, specifically for non-refractory components ($SO_4^{2-}$, $NO_3^-$,
$NH_4^+$, $Cl^-$, and organics). Coggon et al. (2012; 2014) discuss in detail the AMS operational details and results from
some of the campaigns. Cloud water (CW) was collected using a Mohnen CW collector, which was manually placed
above the fuselage of the Twin Otter during cloud penetrations for sample collection into vials kept inside the aircraft.
After flights, samples were analyzed for pH and speciated concentrations of various water-soluble ions and elements,
with a number of studies summarizing the operational details and selected results (e.g., Wang et al., 2014; Wang et
al., 2016; Macdonald et al., 2018). An Oakton Model 110 pH meter was used for E-PEACE, NiCE, and BOAS, and a
Thermo Scientific Orion 8103BNUWP Ross Ultra Semi-Micro pH probe was used for FASE and MACAWS. Water-
soluble ionic composition was measured via Ion Chromatography (IC; Thermo Scientific Dionex ICS – 2100 system),
except some ions during E-PEACE, including Na$^+$, could not be measured. Water-soluble elemental composition was
measured via Inductively Coupled Plasma Mass Spectrometry (ICP-MS; Agilent 7700 Series) for E-PEACE, NiCE,





and BOAS, and via Triple Quadrupole Inductively Coupled Plasma Mass Spectrometry (ICP-QQQ; Agilent 8800
Series) for FASE and MACAWS. Cloud water was not collected during CSM. The IC species analyzed in this study
are Cl⁻, $NH_4^+$, $NO_3^-$, non-sea salt (nss)-$SO_4^{2-}$, and oxalate, and the ICPMS species analyzed are $Ca^{2+}$, $K^+$, $Na^+$, and V.
We used the following equation to calculate nss-$SO_4^{2-}$ under the assumption that all $Na^+$ is from sea salt (e.g.,
AzadiAghdam et al., 2019):

$$[nss - SO_4{}^{2-}] = [SO_4{}^{2-}] - 0.253 \times [Na^+] \tag{1}$$


Aqueous concentrations of ions and elements were converted into air-equivalent concentrations using the mean LWC
encountered when the aircraft was in cloud (LWC > 0.02 g m⁻³) during collection of individual samples.
Aircraft data were analyzed four different ways over the study domain. The primary focus of the analysis is
using data within the spatial domain listed in Sect. 2.1 only when the aircraft was over the ocean. In addition to a LWC
maximum of 0.02 g m⁻³, another screening criterion was utilized to omit data during RFs strongly influenced by
wildfire emissions (Table 3), which was when the median flight-wide $N_{a>10nm}$ value exceeded 7,000 cm⁻³ for altitudes
less than 800 m. This value was determined by closely examining flights that flew through areas with reported wildfire
influence using flight notes. Data were alternatively analyzed for RF segments only over the ocean without the $N_{a>10nm}$
criterion applied, and then also when the aircraft flew within the spatial domain over land and ocean both with and
without the same wildfire criterion; those results are shown in Tables S1 - S3. Note that CSM was the only campaign
for which this criterion was not applied, as smoke was the sole focus of the mission and the flights are considered to
all have been influenced to some extent. Moreover, CSM is unique amongst the campaigns examined where the
scientific hypotheses to be tested are not as applicable due to the widespread smoke coverage, but we still examine it
as it can provide useful insights.
Mann-Whitney U tests were performed for the aircraft data and the CW data, where the null hypotheses ($p \leq$
0.05) were that the medians of certain variables ($N_a$, $N_d$, wind speed and direction) and species concentrations of
southerly and northerly wind days were similar within a campaign.

**2.2      Wind Direction Classification**
To determine boundary layer wind direction in the study region, we used a number of data products, as each
provided unique advantages either related to temporal, spatial, or vertical coverage. Data from NOAA's National Data
Buoy Center (NDBC) were analyzed to verify the ocean surface wind direction was between 135° and 225°, which is
considered southerly in this study. We focused on wind direction during 1400 - 2200 UTC to overlap with when the
majority of RFs occurred (Marina, CA is 7 hours behind UTC). Other days classified as northerly flow adhered to
surface wind direction between 315° and 45°. Five buoys were used to match the ones used in Juliano et al. (2019a):
46011 (Santa Maria: 34.94° N, 120.99° W), 46013 (Bodega Bay: 38.24° N, 123.32° W), 46014 (Point Arena: 39.23°
N, 123.98° W), 46028 (Cape San Martin: 35.77° N, 121.90° W), and 46042 (Monterey: 36.79° N, 122.40° W). Buoy
locations relative to the CA coast are shown in Fig. 1 of Juliano et al. (2019a).
We used Multi-Channel RGB data from the Geostationary Operational Environmental Satellite-WEST Full Disk
Cloud Product (GOES-15) at time resolutions of every three hours for E-PEACE, hourly for NiCE, BOAS, FASE,
and MACAWS, and every half-hour for CSM. Wind direction was assessed via cloud movement, which was partly a
focus of this study (e.g., boundary layer cloud characteristics) with particular attention paid to the principal RF time
period. We investigated all days within a campaign month, and not just days coinciding with a RF. For example, E-
PEACE comprised flights from 9 July to 18 August 2011, and thus GOES data from 1 July through 31 August 2011
were investigated for that year.
The National Oceanic and Atmospheric Administration (NOAA) Hybrid Single-Particle Lagrangian Integrated
Trajectory (HYSPLIT; Stein et al., 2015; Rolph et al., 2017) model was also used to obtain back trajectories based on
North American Mesoscale Forecast System (NAM) meteorological data (12 km resolution) ending at 36.67° N,
121.60° W for 500, 900, 2,500, and 4,500 m AGL. These altitudes were selected to both capture marine boundary
layer (MBL) and free troposphere (FT) winds and reflect the variety of altitudes the Twin Otter aircraft flew at during
the six campaigns in Table 1; however, the trajectories at 500 m were most important for connecting to the aircraft
data analysis.
For Twin Otter flight days, aircraft wind data were used to confirm that wind direction was either southerly or
northerly in the lowest 800 m of the flights (over ocean and land), which was the altitude range of most of the flight





time. For a case-by-case basis, archived surface weather charts were accessed via the NOAA Weather Prediction
Center (WPC) to investigate wind direction at specific sites (like Pt. Reyes).

**2.3    NAAPS and COAMPS**
Both the Navy Aerosol Analysis and Prediction System (NAAPS; Lynch et al., 2016;
https://www.nrlmry.navy.mil/aerosol/) and the Coupled Ocean/Atmosphere Mesoscale Prediction System (COAMPS;
Hodur, 1997) are used to support the analysis of airborne data collected during the six Twin Otter campaigns and
assess how well they can simulate southerly flow on days when observational datasets indicate such flow directions
offshore of CA. NAAPS is a global aerosol forecast model run by the U.S. Naval Research Laboratory (NRL) in
Monterey, CA that predicts 3-dimensional anthropogenic and biogenic fine (ABF), dust, sea salt, and biomass burning
smoke particle concentrations in the atmosphere. NAAPS relies on meteorological data derived from the Navy Global
Environmental Model (NAVGEM; Hogan et al., 2014) and considers 25 vertical levels in the troposphere. For this
study, we utilized the reanalysis version of NAAPS (NAAPS-RA, hereafter called NAAPS) that assimilates aerosol
depth observations to get a general sense of the simulated differences between southerly and northerly flow days for
our region of focus and as a complement to the aircraft data. We investigated data for northward wind speed ($v_{wind}$)
and mass concentrations for ABF aerosols and sea salt (Fig. 2), along with smoke, dust, coarse aerosol, and fine aerosol
(Fig. S2). Note that ABF represents secondarily formed species ($SO_4^{2-}$ and secondary organic aerosol) and primary
organic aerosol generally within the fine mode (<1 µm). To approximate the average boundary layer height of all the
missions used in this study, the first five vertical levels (max height of ~668 m above sea level) of NAAPS were used
for data analysis.
For our analysis, the NAAPS data were first separated into southerly and northerly flow days for each campaign
based on results from Sect. 2.2, and the average value of each parameter was calculated for four reported times: 0000,
0600, 1200, and 1800 UTC. The most focus is placed on 1800 UTC, as that time coincided with most Twin Otter
flight periods (results for the remaining time periods are in Fig. S3-S9). Then, all the parameters except $v_{wind}$ were
summed across the five vertical levels to get a total mass concentration (µg m$^{-3}$) up to ~668 m above sea level, whereas
the average was calculated for $v_{wind}$. Those values were used to calculate the difference between southerly and
northerly flow days at $1.0° \times 1.0°$ spatial resolution.
COAMPS is a high-resolution meteorological forecast model developed by the NRL's Marine Meteorology
Division (MMD) that outputs parameters like air temperature, winds, precipitation, cloud base and top heights, and
mass concentrations for the same aerosol species as those in NAAPS. For this study, we assessed the wind
speed/direction and smoke from COAMPS and NAAPS for the purpose of contrasting with observational data.
COAMPS maps were generated for this study by NRL at three different resolutions: 45 km, 15 km, and 5 km. To
compare to NAAPS, 15 km resolution grids were used. To assess the efficacy of COAMPS and NAAPS at forecasting
heavy pollution on a day with southerly winds, we performed a comparison of the two models for CSM RF 6 at 1800
UTC to match the flight time. The focus areas for both COAMPS and NAAPS matched that of the aircraft data
mentioned in Sect. 2.1.1. The altitudes used for the COAMPS maps for wind speed/direction and smoke were 762 m
and 305 m, respectively, as the best match to the NAAPS maximum altitude used in this work.

**2.4    IMPROVE**
To investigate the difference in surface-level aerosol measurements between southerly and northerly flow days,
this study utilized composition data from the Interagency Monitoring of Protected Visual Environments (IMPROVE)
network (Malm et al., 1994; http://views.cira.colostate.edu/fed/). Data were taken from the Pt. Reyes National
Seashore surface station (38.07° N, 122.88° W) for the full campaign months shown in Table 1. Every third day,
gravimetric mass of particulate matter ($PM_{2.5}$ and $PM_{10}$) was measured. The $PM_{2.5}$ fraction was further analyzed via
ion chromatography and X-ray fluorescence (XRF) for water-soluble ions and elements, respectively, along with
organic and elemental carbon (OC and EC).
This study specifically investigated (µg m$^{-3}$): $PM_{2.5}$, coarse mass ($PM_{coarse} = PM_{10} - PM_{2.5}$), Cl$^-$, NO$_3^-$, SO$_4^{2-}$, Ni,
K$^+$, Si, V, EC, OC, and fine soil. The total OC measurement comes from a summation of four fractions of OC, which
are categorized by a method of carbon analysis detection temperature (e.g., Chow et al., 1993; Watson et al., 1994).
This method quantifies methane produced via volatilization of particulate species in pure helium at 120°C (OC1),
250°C (OC2), 450°C (OC3), and 550°C (OC4). Similarly, the total EC measurement is a summation of three fractions





categorized via combustion temperatures in a 98% pure helium and 2% pure oxygen environment: 550°C (EC1),
700°C (EC2), and 800°C (EC3). Fine soil concentrations are calculated as follows (Malm et al., 1994):
$Fine\ soil\ (\mu g\ m^{-3}) = 2.2 \times [Al] + 2.49 \times [Si] + 1.63 \times [Ca] + 2.42 \times [Fe] + 1.94 \times [Ti]$     (2)
This equation was confirmed by several studies (e.g., Cahill et al., 1981; Pitchford et al., 1981; Malm et al., 1994)
through comparisons of resuspended soils and ambient particles.
Upon examination, it was decided to only use data for E-PEACE and BOAS because those campaign periods
had more than a single point with valid data for southerly days (three and two, respectively); recall that IMPROVE
data are only available every third day so some southerly days would not necessarily have available IMPROVE data.
All the species analyzed had a status flag of "V0" ("Valid value") or "V6" ("Valid value but qualified due to non-
standard sampling conditions"), which are both considered valid data. We chose to include data flagged as "V6" (Cl⁻
, NO₃⁻, and SO₄²⁻ for BOAS) due to the small quantity of usable data for southerly days. Additional information, like
sampling protocols, are provided elsewhere (http://vista.cira.colostate.edu/Improve/sops/). Like the aircraft and CW
data, Mann-Whitney U tests were performed on this dataset to determine if the median species concentrations were
equivalent for southerly and northerly days across a campaign.

### 2.5     MODIS

To assess cloud characteristics of southerly and northerly flow days during the campaign months of this study,
we retrieved daily mean values within the same focus region defined for aircraft data in Sect. 2.1.1 (35.31° N – 40.99°
N, 125.93° W – 118.98° W) for the following properties from the MODerate resolution Imaging Spectroradiometer
(MODIS) on Aqua through NASA Giovanni (https://giovanni.gsfc.nasa.gov/giovanni/): cloud effective particle radius
($r_e$; µm), cloud liquid water path (LWP; g m⁻²), cloud optical thickness (COT), cloud fraction (from cloud mask), and
aerosol optical depth (AOD, combined dark target and deep blue at 0.55 µm for land and ocean). $N_d$ (cm⁻³) was
calculated from MODIS properties based on the following equation (Painemal and Zuidema, 2011):
$N_d = 1.4067 \times 10^{-6}[cm^{-0.5}] \times \frac{COT^{0.5}}{r_e^{2.5}}$     (3)
Additionally, retrieval data were only used when cloud fraction ≥ 30% to maximize both data reliability and sample
size (Mardi et al., 2021). The focus of the analysis is comparing median values of these remotely sensed variables
between southerly and northerly days for E-PEACE and BOAS due to a similar LWP value for the two flow regimes
(66.48/67.17 g m⁻² and 84.40/89.90 g m⁻², respectively). Data for the other campaigns are included in the SI.
Additionally, this study used MODIS visible imagery on NASA Worldview to qualitatively identify smoke plumes,
in addition to fire radiative power from the MODIS Fire Information for Resource Management System (FIRMS;
https://earthdata.nasa.gov/firms).

### 3     Results and Discussion
### 3.1     Lower Tropospheric Wind Profile

We first examine NAAPS and airborne observations for the lower tropospheric wind profile during the
periods of analysis shown in Table 1. Note that the other datasets described in Sect. 2.2 are consistent with the airborne
wind results and thus only NAAPS and aircraft data are discussed here for two reasons: NAAPS results are used to
assess how such a model quantifies differences in winds between southerly and northerly flow days as identified with
methods in Sect. 2.2, whereas aircraft data provide insight into typical wind speeds during southerly and northerly
flow periods.
Beginning with the aircraft data, results are discussed here only for measurements over the ocean with the
$N_{a>10nm}$ filter applied to remove smoke influence (Table 3). The mode of wind directions during southerly and northerly
flow days in each campaign expectedly aligned with southerly (144° – 194°) and northerly flow (327° – 332°),
respectively, because of how the classification was done (Sect. 2.2). Median wind speeds across each campaign ranged
from 2.35 – 7.75 m s⁻¹ for southerly flow in contrast to 5.12 – 8.87 m s⁻¹ for northerly flow. This finding differs from
what has been observed in previous studies, likely due to the difference in sampling location: aircraft observations
from the surface to 800 m versus buoy/surface observations, respectively. All campaigns featured higher median wind
speeds for northerly flow flights. Both the median wind speeds and directions of southerly and northerly days were
significantly distinct from one another for all of the studied campaigns (Table S4).





**Table 3: Median values (southerly/northerly) of various parameters over the ocean with an $N_{a>10nm}$ filter such that RFs with median $N_{a>10nm} > 7,000$ cm$^{-3}$ were removed from the final analysis to eliminate smoke interference. Mode values are used for wind direction. The instruments used for the parameters from left to right are as follows: CPC 3010, CPC 3010 – PCASP$_{<1\mu m}$, PCASP$_{<1\mu m}$, PCASP$_{>1\mu m}$, CPC 3025/CPC 3010, CASF. The far right-hand columns indicate the number of datapoints used from each campaign, with $n_{Na}$ indicating the amount of data used for all $N_a$ calculations, $n_{Nd}$ is for cloud data, and $n_{Wind}$ is for wind speed and direction. FSSP data were used for $N_d$ data only during BOAS, whereas CASF was used in other campaigns. These data are for the lowest 800 m above sea level. The reader is referred to Fig. S10 for box plots corresponding to the analysis in this table, as well as Table S4 for Mann-Whitney U p-values.**

| | $N_{a>10nm}$ (cm$^{-3}$) | $N_{a10-100nm}$ (cm$^{-3}$) | $N_{a0.1-1\mu m}$ (cm$^{-3}$) | $N_{a>1\mu m}$ (cm$^{-3}$) | $N_{a3}:N_{a10}$ (-) | $N_d$ (cm$^{-3}$) | Wind Speed (m s$^{-1}$) | Wind Direction (°) | $n_{Na}$ (×10$^3$) | $n_{Nd}$ (×10$^3$) | $n_{Wind}$ (×10$^3$) |
|---|---|---|---|---|---|---|---|---|---|---|---|
| E-PEACE | 861 / 703 | 501 / 454 | 338 / 197 | 0 / 1.25 | 1.09 / 1.10 | 252 / 163 | 3.38 / 7.58 | 177.61 / 330.48 | 20.3 / 202.7 | 17.1 / 127.1 | 37.4 / 330.8 |
| NiCE | 953 / 606 | 248 / 245 | 471 / 260 | 2.51 / 0 | 1.12 / 1.17 | 249 / 254 | 3.80 / 5.12 | 180.81 / 327.20 | 1.4 / 66.8 | 1.5 / 39.6 | 3.0 / 112.8 |
| BOAS | 750 / 497 | 553 / 256 | 204 / 196 | 0 / 1.24 | 1.20 / 1.18 | 143 / 127 | 5.49 / 6.35 | 166.97 / 328.58 | 5.8 / 72.1 | 3.9 / 20.5 | 11.8 / 104.7 |
| FASE | 836 / 916 | 423 / 635 | 326 / 180 | 0 / 0 | 1.29 / 1.16 | 203 / 223 | 2.35 / 6.82 | 144.03 / 331.29 | 1.0 / 95.5 | 0.3 / 99.2 | 1.3 / 194.9 |
| MACAWS | 722 / 815 | 560 / 635 | 154 / 164 | 0 / 0 | 1.25 / 1.26 | 189 / 165 | 7.75 / 8.87 | 162.15 / 330.28 | 10.3 / 118.9 | 6.6 / 27.0 | 16.9 / 145.9 |
| CSM | 5,558 / 3,451 | 5,081 / 3,366 | 515 / 365 | 1.00 / 0 | 1.30 / 1.67 | 334 / 314 | 6.10 / 6.77 | 193.93 / 332.16 | 4.8 / 31.5 | 1.8 / 4.1 | 6.9 / 41.3 |

NAAPS values are discussed for $v_{wind}$ for the lowest ~668 m above sea level, with positive (negative) values representing southerly (northerly) flow (Fig. 2). This altitude range coincides with the airborne data shown in Table 3. The $v_{wind}$ data are categorized into "Southerly Days," "Northerly Days," and "Southerly – Northerly Days" for 1800 UTC, which overlaps with most of the Twin Otter flight times (Fig. 1); results for 0000, 0600 and 1200 UTC are provided in Fig. S3. Both southerly and northerly days had less negative $v_{wind}$ closer to the coast (up to 35° N) compared to farther offshore over the ocean (~ -3/-9 and -4/-6 m s$^{-1}$, respectively, for southerly/northerly flow). Slower, slightly northerly winds extended farther north to Marina and west to 123.5° W for southerly days, which is illustrated in red (differences exceeding ~3 m s$^{-1}$ between flow regimes) in the "Southerly – Northerly Days" panel. Northerly days also had an area of less negative $v_{wind}$ north of 43.5° N, which is emphasized in the "Southerly – Northerly Days" panel in blue (differences of -4 – -6 m s$^{-1}$). NAAPS was not able to fully capture southerly winds over the ocean and along the coast in that $v_{wind}$ was not clearly positive; however, the magnitude of the wind speed difference along the coastal area of the study domain appeared to align with the mechanics of coastal wind reversal and CTDs: the weakening of northerly wind and ultimate reversal of flow (e.g., Winant et al., 1987; Melton et al., 2009). A key conclusion from NAAPS is that the difference between southerly and northerly flow days matches expectations with southerly days having at least a greater tendency towards more positive $v_{wind}$ but still not necessarily distinctly positive.



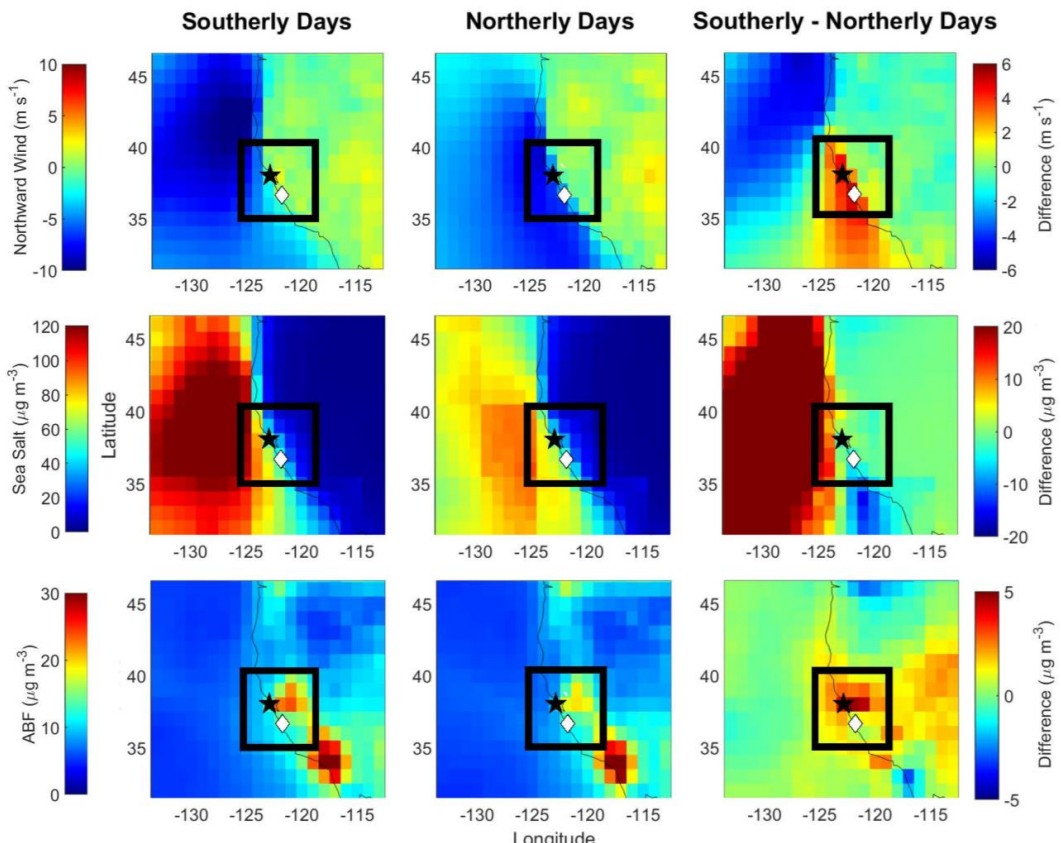

**Figure 2: Average northward wind speed ($v_{wind}$; m s$^{-1}$), total sea salt mass concentration (µg m$^{-3}$), and total ABF mass concentration (µg m$^{-3}$) of campaign months at 1800 UTC for 1st through 5th NAAPS levels (up to ~668 m above sea level) for southerly and northerly flow wind days. The right-most panel illustrates the difference between southerly and northerly flow days. The airbase in Marina, CA is denoted by a white diamond, Pt. Reyes is indicated with a black star, and the black box indicates the region of focus in this study.**

### 3.2 Aerosol Response to Southerly Flow

#### 3.2.1 Fire Radiative Power Maps

Prior to discussing aerosol results, we address the influence of wildfire emissions, which is an aerosol source that varies in terms of strength between the six campaign periods in contrast to shipping and other forms of continental emissions that are more consistent year to year. Past studies using airborne and surface-based data at Marina overlapping with the six campaigns in Table 1 revealed the following in terms of notable biomass burning influence around Marina and offshore areas (e.g., Prabhakar et al., 2014; Braun et al., 2017; Mardi et al., 2018): (i) E-PEACE/BOAS: no major influence of note; (ii) NiCE: influence around the last week of July 2013; (iii) FASE: influence between 25 July and 12 August; (iv) MACAWS: significant influence on flights during 28 June and 3 July owing to the aircraft having flown close to wildfire areas inland in northern CA; (v) CSM: significant influence throughout the campaign. These archived notes do not preclude the possibility of biomass burning influence during other periods of those campaigns as it relates to Twin Otter aerosol and cloud measurements.

Spatial maps of fire radiative power (FRP; Fig. 3), indicative of burn intensity, show relatively less burning activity in immediate proximity to Marina during E-PEACE and BOAS. In contrast, the other campaigns show clusters of burning spots around Marina. Note that CSM, by virtue of its name, was focused largely on wildfires with dedicated RFs to sample smoke. MACAWS also was designed as a wildfire study but had less cases of strong plumes to sample,



which included RFs on 28-29 June farther inland than most RFs, resulting in very high aerosol number concentrations
($N_{a>10nm} > 10,000$ cm$^{-3}$).
**3.2.2    Fine Aerosol**
The first hypothesis of this study is that southerly flow yields higher fine aerosol levels associated with
anthropogenic and continental tracer species due to more perceived influence from land and shipping sources (Juliano
et al., 2019a; 2019b). This was also speculated by Hegg et al. (2008) although it was not examined in great detail by
that study. Here we rely on results from a number of datasets including measurements from the Twin Otter (Tables 3
and 4) and the Pt. Reyes IMPROVE site (Fig. 4), along with NAAPS model results (Fig. 2).
**3.2.2.1      Airborne: Particle Concentration**
Beginning with the Twin Otter data, aerosol data for 17 southerly flight days corresponding to 21 RFs were
compared to 93 other flight days with predominantly northerly flow in Table 3 (box plots of the variables in Fig. S10,
and Mann-Whitney U test results are in Table S4), as well as Tables S1-S3. We focus primarily on flight data over the
ocean with the $N_{a>10nm}$ filter applied to omit wildfire influence; the other aircraft data result tables in the Supplement
generally show the same trends as Table 3. We caution that the results of FASE, and to a slightly lesser extent NiCE,
are not as meaningful as the other campaigns owing to the least amount of statistics for southerly conditions, with
numbers of datapoints shown in the tables.
The total submicron aerosol number concentration, $N_{a>10nm}$, was far larger for southerly flow (722-5,558 cm$^{-3}$)
as compared to northerly flow flights (497-3,451 cm$^{-3}$). Of the six campaigns, the only ones with higher median
values in northerly flow were FASE and MACAWS, with small $\Delta N_{a>10nm}$ of -80 cm$^{-3}$ and -93 cm$^{-3}$, respectively. CSM
exhibited the largest difference in median values for $N_{a>10nm}$ between southerly and northerly flow ($\Delta N_{a>10nm} = 2,107$
cm$^{-3}$), followed by NiCE ($\Delta N_{a>10nm} = 347$ cm$^{-3}$) and BOAS ($\Delta N_{a>10nm} = 253$ cm$^{-3}$). While these campaigns have a
smaller relative sample size of southerly data ($n_{Na} < 6 \times 10^3$; CSM: $4.8 \times 10^3$, NiCE: $1.4 \times 10^3$; and BOAS: $5.8 \times 10^3$), E-
PEACE has a sizable amount of southerly data ($20.3 \times 10^3$) and the least fire influence of the missions included in this
study, so we find it may be the most reliable campaign to analyze. There was a distinct difference between southerly
and northerly days during E-PEACE as well, with a $\Delta N_{a>10nm}$ of 158 cm$^{-3}$. As the number concentration in the
submicron range dominates the total CPC concentrations, these results convincingly point to an enhancement of fine
aerosol pollution in southerly flow even without the $N_{a>10nm}$ filter (Table S1).
We examined various size ranges of particles in the submicron range as well. For particles between 10-100
nm, southerly conditions generally had higher number concentrations except again for FASE and MACAWS and with
more comparable levels during NiCE. As particles larger than 100 nm are more relevant for cloud condensation nuclei
(CCN) activity, we also examined number concentrations for diameters between 0.1 and 1 µm, which show higher
southerly levels except for MACAWS. Between campaigns, CSM overall exhibited the highest particle concentrations
in this size range due to extensive wildfire emissions in the area, which are known to be linked with enhanced levels
of particles larger than 100 nm in the same region (Mardi et al., 2018), which is why this campaign shows relatively
large PCASP enhancements in both southerly and northerly flow conditions relative to the other campaigns (see in
particular Tables S1-S2). Without the CPC filter (Table S1), only the medians for NiCE and BOAS on northerly wind
days changed, resulting in the $N_{a10-100nm}$ median during NiCE to be lower during southerly flow days compared to
northerly days. When looking within the region of focus, the inclusion of land data in addition to ocean data (Tables
S2-S3) leads to significant $N_a$ differences (to a lesser extent for the filtered data, Table S3) compared to Table 3,
including higher submicron concentrations for NiCE, BOAS, and FASE.



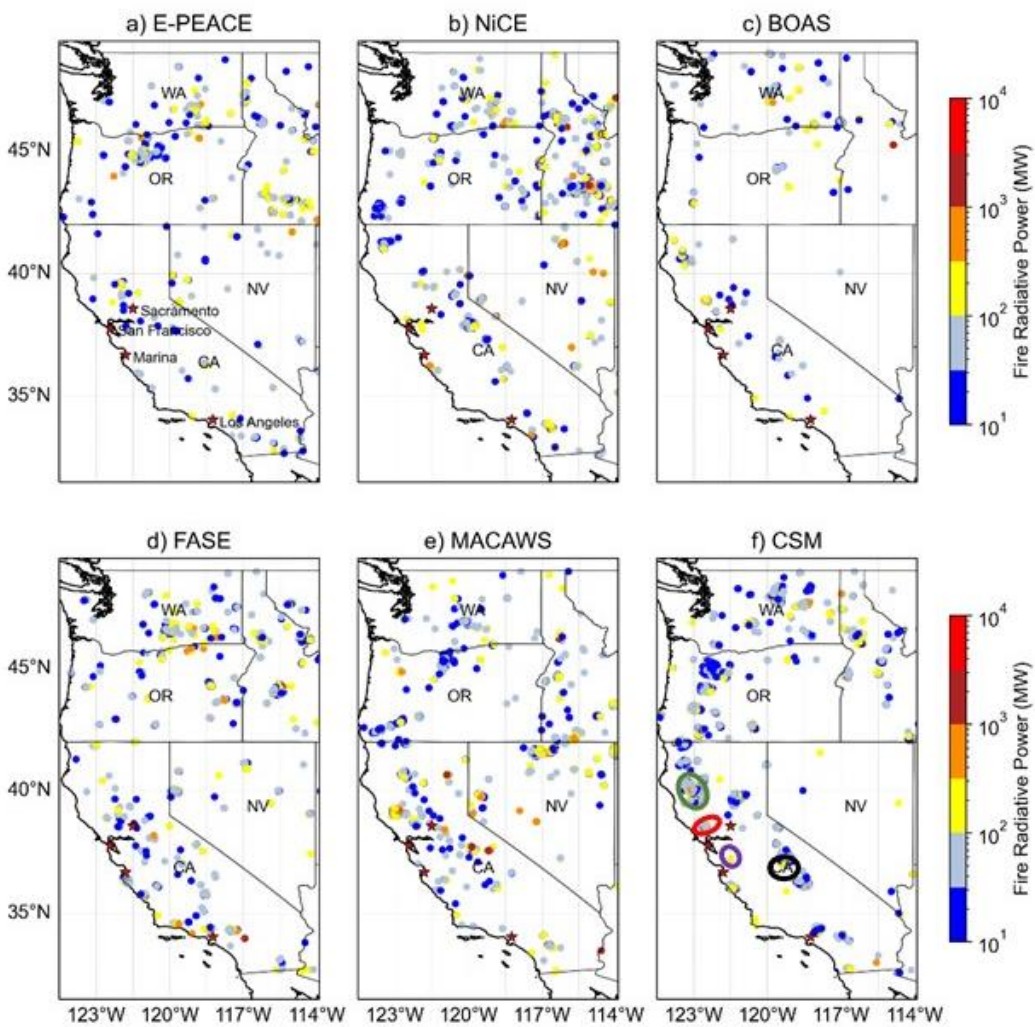

**Figure 3: Spatial maps of fire radiative power (FRP), downloaded from the MODIS Fire Information for Resource**
**Management System (FIRMS; https://earthdata.nasa.gov/firms) for the entire months spanning individual field**
**campaigns in Table 1. Only FRP values with a high detection confidence level (≥ 80%) are shown (Giglio et al., 2015). The**
**circled areas in panel (f) correspond to some of the largest wildfires in CA state history that occurred in 2020 that are**
**referred to in Sect. 3.4.2: August Complex fire (green), SCU Lightning Fire Complex (purple), Creek fire (black), and**
**LNU Lightning Complex fire (red).**
Although new particle formation (NPF) was not expected to be prominent in the lower 800 m owing mostly
to high aerosol surface areas especially due to sea spray emissions, we still examined the ratio of $N_a$ above 3 nm
relative to 10 nm ($N_{a3}$:$N_{a10}$), as this ratio is a commonly used marker for identifying NPF. Such instances are more
common in the free troposphere in the study region owing to reduced aerosol surface areas (Dadashazar et al., 2019).
The results suggest that the $N_{a3}$:$N_{a10}$ ratios for the two flow regimes were significantly different for all the campaigns
except for MACAWS (higher ratios in southerly flow for BOAS and FASE), with median flow direction-dependent



values per campaign ranging from 1.09 to 1.30. During CSM, the median ratio value was 1.67 in northerly flow
conditions due to presumed influence from high precursor levels in smoke plumes.

### 3.2.2.2      Airborne: Tracer Species in Cloud Water

We next turn to CW composition data (Table 4) to continue learning more about the effect of southerly flow
and its associated emission sources. NiCE and FASE were not included in the CW calculations of Table 4 (but shown
in Fig. S11) because there were fewer than five samples from RFs with southerly wind direction for those two
campaigns, and CW was not collected during CSM. $NO_3^-$ and nss-$SO_4^{2-}$, both representative of fine aerosol pollution,
were higher for southerly days, with a significant difference (Table S5) apparent in E-PEACE (1.80/0.30 and 2.10/0.81
$\mu$g m$^{-3}$ for southerly and northerly days, respectively), as well as for $NO_3^-$ during BOAS (1.02/0.23 $\mu$g m$^{-3}$ for southerly
and northerly days, respectively). The same trend was observed for V (ship exhaust tracer) and $NH_4^+$, which can be
used as a tracer for continental sources such as agriculture (Juliano et al., 2019b). Thus, these results help to provide
more confidence in results from Juliano et al. (2019b) but with increased statistics across more campaigns. For E-
PEACE and MACAWS, there were also lower southerly flow concentrations of $K^+$ (0.01/0.05 and 0.06/0.11 $\mu$g m$^{-3}$)
and $Ca^{2+}$ (0.05/0.07 and 0.06/0.16 $\mu$g m$^{-3}$), suggestive of less influence from biomass burning and dust sources with
the caveat that $K^+$ and $Ca^{2+}$ have sources other than biomass burning and dust.
There were also higher concentrations of oxalate during southerly days, which can be used as a tracer for aqueous
processing (Hilario et al., 2021), wherein cloud droplets are formed from oxidized volatile organic compounds (Ervens
et al., 2011; Ervens, 2015; Mcneill, 2015). Further, there were significant differences in median concentrations
between southerly and northerly flow days during BOAS and MACAWS (0.12/0.05 and 0.08/0.03 $\mu$g m$^{-3}$,
respectively). Precursors to oxalate are diverse including from biogenic sources, biomass burning, combustion (e.g.,
Stahl et al., 2020 and references therein), shipping, along with being associated with sea salt and dust owing to gas-
particle partitioning (Sorooshian et al., 2013; Stahl et al., 2020; Hilario et al., 2021); such sources are presumed to be
influential during southerly flow based on the notion that air masses are influenced by some combination of continental
emissions and extended time in shipping lanes.
Cloud water pH was lower and thus more acidic on southerly days for all three campaigns (3.85/4.54, 4.30/4.34,
4.33/4.62 for southerly/northerly days during E-PEACE, BOAS, and MACAWS, respectively, and statistically
different for E-PEACE and BOAS), which is another indicator for anthropogenic pollution enriched with acidic
species (Pye et al., 2020). Increased acid levels can result in more $Cl^-$ depletion when considering sea salt particles
(e.g., Edwards et al., 2023 and references therein); interestingly, southerly days were characterized by lower $Cl^-$:$Na^+$
ratios with median values of 1.39 (MACAWS), 1.63 (E-PEACE) (both campaigns of which southerly days were
significantly different from northerly flow days), and 2.48 (BOAS), although the difference in MACAWS was only
0.12. Braun et al. (2017) noted that, theoretically, over 60% of the $Cl^-$ depletion in the submicron range could be
attributed to nss-$SO_4^{2-}$, and greater than 20% in the supermicron range could be attributed to $NO_3^-$. As was noted
previously, nss-$SO_4^{2-}$ and $NO_3^-$ were noticeably enhanced during southerly flow days while the $Cl^-$:$Na^+$ ratios were
reduced. Schlosser et al. (2017) also reported that organic acids, notably oxalate, were significantly enhanced during
periods of $Cl^-$ depletion, which is reflected in our CW data. As E-PEACE was statistically the most robust dataset (and
all CW species except $Ca^{2+}$, $NH_4^+$, and oxalate had medians that were significantly different between southerly and
northerly flow days), the results from CW convincingly align with more shipping and/or continental influence in
southerly flow to impact cloud composition.

**Table 4: Median values (southerly/northerly) of water-soluble CW composition ($\mu$g m$^{-3}$) over the entirety of three campaigns with sufficient data. The starred (\*) values are reported in ng m$^{-3}$. The number of samples used in each campaign is in the far-right hand column (n). The reader is referred to Table S5 which shows the p-values from the Mann-Whitney U tests, as well as Fig. S11 which shows box plots of the CW composition results for the five campaigns with available data. Values shown as "–" denote when samples were below the limit of detection.**

| | $Ca^{2+}$ | $Cl^-$/$Na^+$ | $K^+$ | $Na^+$ | $NH_4^+$ | $NO_3^-$ | Oxalate | pH | nss-$SO_4^{2-}$ | V | n |
|---|---|---|---|---|---|---|---|---|---|---|---|
| E-PEACE | 0.05/0.07 | 1.63/2.15 | 0.01/0.05 | 0.42/1.21 | —/— | 1.80/0.30 | 0.02/0.02 | 3.85/4.54 | 2.10/0.81 | 2.16\*/0.38\* | 10/65 |
| BOAS | 0.11/0.08 | 2.48/2.74 | 0.06/0.06 | 1.99/1.55 | 0.44/0.04 | 1.02/0.23 | 0.12/0.05 | 4.30/4.34 | 1.08/0.83 | —/0.15\* | 5/21 |
| MACAWS | 0.06/0.16 | 1.39/1.51 | 0.06/0.11 | 1.30/2.70 | 0.08/0.05 | 0.55/0.38 | 0.08/0.03 | 4.33/4.62 | 0.56/0.26 | 0.07\*/0.05\* | 15/51 |

### 3.2.2.3 Surface: Aerosol Composition

We next examine surface composition data from the Pt. Reyes IMPROVE site. Mass concentrations of twelve PM composition variables were investigated to analyze important tracers along the coast (Fig. 4), with Mann-Whitney U test p-values for comparing southerly and northerly flow days shown in Table S6. It is important to recall that E-PEACE and BOAS were the only campaigns that had more than a single day of valid data coinciding with southerly flow because of the added challenge of IMPROVE sampling occurring every third day; therefore, northerly days had significantly more data points (18 for E-PEACE and seven for BOAS) compared to southerly days (three and two, respectively). That is the general reason for the large whiskers on the box plots for northerly RFs during E-PEACE and the lack of whiskers for southerly RFs during BOAS. Another feature to note is the 'folded over' appearance of some of the box plots. This indicates a high variance within the dataset and a skewed distribution. We caution that this analysis is not very statistically robust owing to the rare nature of southerly days in overlap with IMPROVE sampling; however, we take a 'better than nothing' approach to use in a supportive role in comparison to other datasets used to assess differences between southerly and northerly flow.

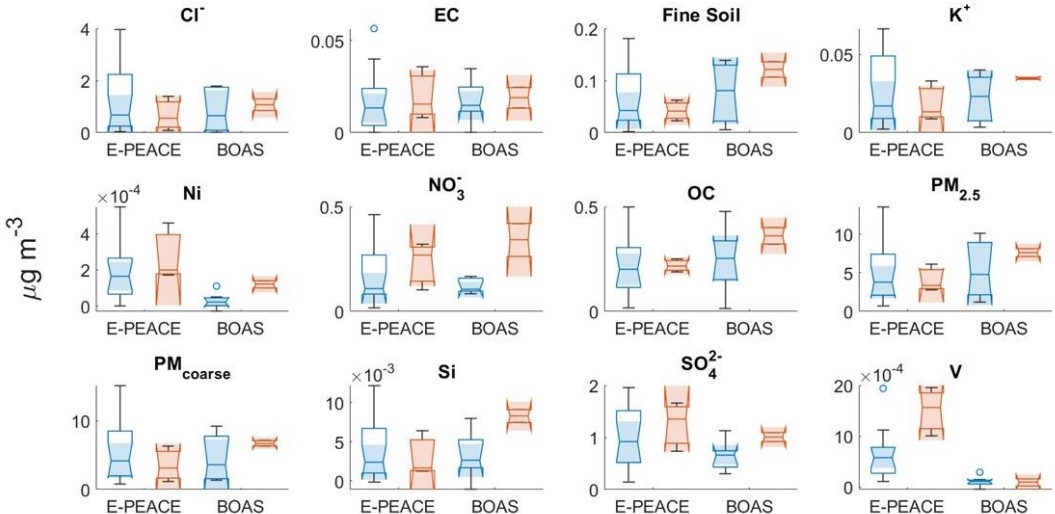

**Figure 4: Box plots of IMPROVE data from the Pt. Reyes surface station. The southerly data for E-PEACE and BOAS (three and two points, respectively) are represented by the red boxes, and the northerly data (18 and seven, respectively) are represented by the blue boxes.**

$SO_4^{2-}$, $NO_3^-$, OC, V, Ni, and EC are reasonable tracer species representative of either shipping and/or continental sources in the study region (Wang et al., 2014; Maudlin et al., 2015; Wang et al., 2016; Dadashazar et al., 2019; Ma et al., 2019) hypothesized to be more enhanced in the coastal CA zone on southerly flow days due to air spending time over shipping lanes and land upwind of the study region. Even with the limited southerly flow statistics, the results of Fig. 4 support this idea as southerly conditions coincide with higher median concentrations of these species than northerly days. The most striking relative differences were for $NO_3^-$ (southerly/northerly): 0.27/0.11 and 0.34/0.10 µg m$^{-3}$ for E-PEACE and BOAS, respectively. $NO_3^-$ was the only species during BOAS that was found to have a median concentration that was statistically different between southerly and northerly days (Table S6). Ni and V are the primary trace metals in heavy ship fuel oils and are commonly used as tracers for ship emissions (Celo et al., 2015; Corbin et al., 2018), and V was previously found enhanced in CW linked to ship emissions in E-PEACE (Coggon et al., 2012; Prabhakar et al., 2014). There were mostly higher concentrations of these species on southerly flow days (E-PEACE southerly/northerly: 0.20/0.17 and 1.56/0.58 ng m$^{-3}$, respectively; BOAS southerly/northerly: 0.12/0.02 and 0.09/0.11 ng m$^{-3}$, respectively), supporting the hypothesis of elevated shipping emissions. Also, a Mann-Whitney U test found that the median V concentrations during E-PEACE were statistically different for southerly and northerly days (Table S6).





Only BOAS exhibited higher $PM_{2.5}$ during southerly days compared to northerly days (7.61/4.82 µg m$^{-3}$,
respectively), with E-PEACE having roughly equivalent concentrations for the two flow regimes (3.39/3.78 µg m$^{-3}$,
respectively). This is likely owing to how $PM_{2.5}$ is not the best marker for shipping and continental emissions owing
to its inclusion of other species of marine and natural origin.

### 3.2.2.4     NAAPS: Aerosol Composition

To round out discussion of fine aerosol pollution, we discuss NAAPS model results (Fig. 2). The largest
enhancements in ABF mass concentrations occurred inland both north of Marina around Pt. Reyes and near the Ports
of Los Angeles and Long Beach. There was >5 µg m$^{-3}$ difference in ABF concentration between southerly and
northerly days near Pt. Reyes. This suggests that while there were elevated levels of anthropogenic emissions in this
area regardless of the flow regime, there were increased concentrations during southerly flow days according to
NAAPS. Additionally, there is a strong ABF signal (>30 µg m$^{-3}$) around 34° N, 118° W for both categories of days,
which is close to the Ports of Los Angeles and Long Beach, two of the busiest container ports (in terms of cargo
volume processed) in the United States and areas with elevated levels of $NO_x$ and $SO_x$ due to the ship exhaust and
port emissions (Corbett and Fischbeck, 1997). As can be seen in the Fig. S5, the ABF concentrations around 34° N,
118° W and 38° N, 122° W increase throughout the day, with more significant increases north of the ports for southerly
flow days. On southerly flow days, NAAPS results point to marked enhancements in fine aerosol and smoke mass
concentration north of Pt. Reyes over water but with mostly a reduction in such values to the south of Pt. Reyes over
water. ABF represents the category of species that are most tied to the tracer species shown already to be enhanced in
southerly flow, and thus at least this result from NAAPS is consistent with enhanced values across most of the study
domain in southerly flow.

### 3.2.3     Supermicron Aerosol

While this study hypothesizes that most of the aerosol changes in southerly flow will pertain to submicron
aerosol, we still discuss supermicron aerosol characteristics to determine if there was any change observed. Beginning
with the aircraft observations, $N_{a>1µm}$ levels were generally low and usually zero in terms of flight median values
simply due to so many zero values during a RF. Northerly flow conditions yielded median levels exceeding zero for
E-PEACE (1.25 cm$^{-3}$) and BOAS (1.24 cm$^{-3}$). In contrast, southerly flow led to levels of 2.51 cm$^{-3}$ and 1.00 cm$^{-3}$
during NiCE and CSM, respectively. The enhancement during southerly flow during at least CSM is presumed to be
due to pervasive smoke during many of those RFs. Figure S1 shows a scatterplot of total CASF number concentration
versus effective diameter to separate out where cloud droplets are relative to probable sea salt particles and then coarse
aerosol associated with the wildfires. There is considerable data coverage at LWC < 0.02 g m$^{-3}$, with effective
diameters below 5 µm and number concentrations exceeding 10 cm$^{-3}$, with the latter surpassing what would be
expected from sea salt (e.g., Gonzalez et al., 2022). It is very likely that dust particles can be entrained into regional
smoke plumes as discussed in past work for the region (e.g., Maudlin et al., 2015; Schlosser et al., 2017). This will be
discussed in more detail for a case flight demonstrating such high levels during southerly flow in Sect. 3.4.2.
Airborne CW results reveal generally no strong trends in either sea salt or dust tracer species between the flow
regimes. The sea salt tracer species $Na^+$ was lower for southerly days during E-PEACE (and statistically different)
and MACAWS (0.42/1.21 and 1.30/2.70 µg m$^{-3}$ for southerly/northerly days) but with an increase during BOAS (1.99
versus 1.55 µg m$^{-3}$). The dust tracer species $Ca^{2+}$ was, expectedly, much less abundant compared to $Na^+$, without
significant differences between flow regimes. However, as already noted (Sect. 3.2.2.2), the fine pollution in southerly
flow likely still influenced supermicron aerosol characteristics via $Cl^-$ depletion in salt particles.
In terms of IMPROVE data, $PM_{coarse}$, Si, fine soil, and $Cl^-$ are the variables that would best coincide with
typical sources of supermicron aerosol (i.e., dust and sea salt). They did not reveal any consistent trend for the two
campaigns. Based on the lack of a general trend and limited southerly statistics, it is concluded that there is insufficient
evidence from IMPROVE to conclude that there is more or less dust or salt influence on southerly days.
The wind profile discussed in Sect. 3.1 has implications for sea salt aerosol production, which is influenced by
wind speed. The breaking of wave crests to produce (mostly coarse mode) spray droplets occurs at strong wind
conditions (>10 m s$^{-1}$) (Monahan et al., 1986). Additionally, jet droplets are produced via bubble bursting at lower
wind speeds (>5 m s$^{-1}$; Blanchard and Woodcock, 1957; Fitzgerald, 1991; Wu, 1992; Moorthy and Satheesh, 2000).
On southerly days, there were faster northerly winds over the open ocean offshore west of 125° W, which
corresponded to high sea salt concentrations (>100 µg m$^{-3}$) according to NAAPS, whereas northerly days had slower



$v_{wind}$ and less sea salt (65 – 90 µg m$^{-3}$) in those same areas farther offshore. In contrast, in the coastal areas south of
35° N, northerly days had higher sea salt concentrations (by 10 – 20 µg m$^{-3}$) than southerly days with weaker (less
negative) $v_{wind}$. NAAPS shows the same general trends for coarse aerosol mass compared to sea salt, with dust being
far less abundant and more spatially heterogeneous in terms of enhancements and reductions between southerly and
northerly conditions. In general, the NAAPS results are consistent with aircraft and IMPROVE results in that in the
study domain, there was not any pronounced difference in coarse aerosol characteristics during southerly flow.
**3.3    Cloud Responses**
**3.3.1    Airborne In Situ Results**
As most campaigns exhibited higher $N_a$ on southerly flight days, it matches expectation that most campaigns
exhibited higher $N_d$ values for southerly days (southerly/northerly values): E-PEACE (252/163 cm$^{-3}$), BOAS (143/127
cm$^{-3}$), MACAWS (189/165 cm$^{-3}$), and CSM (334/314 cm$^{-3}$). These campaigns had southerly $N_d$ values that were ~
20±4 cm$^{-3}$ greater than the median values on northerly days, with a significant difference during E-PEACE ($\Delta N_d$ ~
89 cm$^{-3}$). E-PEACE also had the best cloud data statistics compared to the other missions, qualifying it as the most
robust campaign for inspection of cloud properties. The remaining two campaigns had the least amount of cloud data
during southerly flow conditions (NiCE and FASE) and thus those results are of less importance to discuss. CSM had
the highest $N_d$ concentrations for both southerly and northerly days due to the strongest levels of pollution (from
smoke) relative to the other campaigns.
**3.3.2    Satellite Data Results**
The second part of our hypothesis was that there would be a noticeable difference in cloud properties like $N_d$, $r_e$,
and COT between southerly and northerly flow days (at fixed LWP), namely due to the change in emissions sources.
In particular, we anticipated higher $N_d$ and COT and lower $r_e$ for southerly flow periods due to the Twomey effect
(Twomey, 1974) and higher particle concentrations from continental pollution and shipping emissions. Six parameters
were retrieved from MODIS, divided into southerly and northerly days for E-PEACE and BOAS, and visualized as
box plots (Fig. 5). Cloud LWP medians for southerly and northerly days within E-PEACE (66.48/67.17 g m$^{-2}$) and
BOAS (84.40/89.90 g m$^{-2}$) were not significantly different. Therefore, these two campaigns are the focus here, unlike
the other campaigns that had larger differences (Table S7). The medians for $N_d$ were higher for southerly days
(138.54/91.99 cm$^{-3}$ and 96.59/72.80 cm$^{-3}$ for southerly/northerly wind days during E-PEACE and BOAS,
respectively), and the southerly and northerly medians during E-PEACE were significantly different from one another.
Consistent with the Twomey effect (Twomey, 1974), the median $r_e$ for southerly flow days was lower than northerly
flow days (9.94/11.97 µm and 11.77/13.29 µm), with the medians during E-PEACE being significantly different.
Cloud optical thickness was also higher for southerly days compared to northerly days for both campaigns (10.27/8.42
and 11.88/10.87 for E-PEACE and BOAS, respectively); however, the medians for each flow regime were not found
to be significantly different from one another. We note that even NiCE with LWP values being slightly higher for
southerly days (82.78 g m$^{-2}$ versus 74.54 g m$^{-2}$), the same general results are observed with southerly days having
higher $N_d$/COT and reduced $r_e$ (Table S7); the other three campaigns did not follow these $N_d$/COT/$r_e$ trends due to the
larger LWP differences between flow regimes.
Although no differences were necessarily expected, we still examined cloud fraction and AOD, which were
similar within a campaign for the two types of days (0.47/0.44 versus 0.58/0.57, and 0.10/0.09 versus 0.12/0.11,
respectively, for southerly and northerly wind days during E-PEACE versus BOAS). Based on these results, $N_d$, $r_e$,
and COT differences between flow regimes match our hypothesis, and two out of the three parameters during E-
PEACE were found to be significantly different between southerly and northerly days.



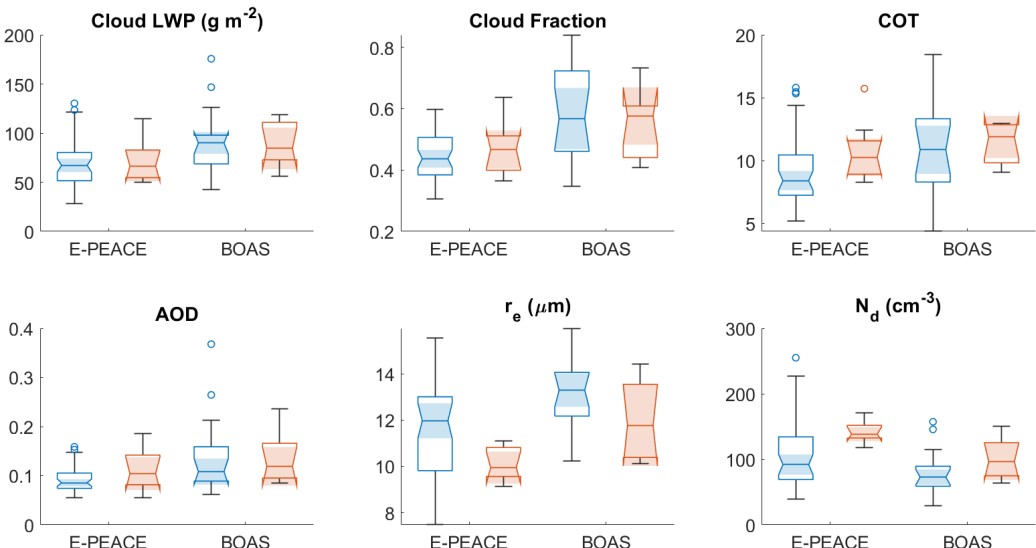

**Figure 5: Box plots of MODIS data within the study region during the periods overlapping with E-PEACE and BOAS. The**
**southerly data for E-PEACE and BOAS (eight points each) are represented by the red boxes, and the northerly data (44**
**and 17 points, respectively) are represented by the blue boxes. The notches (and shading, which helps to more clearly**
**indicate where the notches end) of the boxes assist in the determination of significance between multiple medians. If the**
**notches overlap, the medians are not significantly different from one another.**

### 3.4    Case Studies

In addition to looking at whole campaigns, we also looked closely at two RFs with southerly wind direction:
NiCE RF 16 (29 July 2013) and CSM RF 6 (10 September 2020). NiCE RF 16 was a unique flight, which coincided
with a CTD event (Bond et al., 1996; Nuss, 2007) and its flight path extended past 125° W into a large stratocumulus
cloud clearing (Crosbie et al., 2016; Dadashazar et al., 2020), which was unusual for the Twin Otter flights. CSM RF
6 was on a heavily polluted day owing to biomass burning emissions during one of the worst wildfire periods in CA
history.

#### 3.4.1    NiCE Research Flight 16

NiCE RF 16 (29 July 2013) occurred on a day with a large stratocumulus cloud deck clearing, which, at its
widest point, was 150 km (Crosbie et al., 2016). As noted in Crosbie et al. (2016), this was a CTD event during the
time of the flight, and the boundary layer wind reversal (and resulting northwesterly flow) occurred under the
stratocumulus cloud deck within 100 km of the coast (~ 36.7° N, 123° W). The location of the wind reversal was
known, which allowed us to investigate if there was any apparent gradient in aerosol and cloud variables from the
coast to out over the ocean. The aircraft departed from Marina at approximately 1700 UTC, with a nearly straight,
westward path (Fig. 6a) toward the clear-cloudy boundary (reader is referred to Fig. 1a of Crosbie et al., 2016 for
boundary location). At the clear-cloudy interface (~ 36.7° N, 125° W, 1845 – 2000 UTC), stacked legs were performed
at multiple levels in both the MBL and FT on both sides of the boundary. Subsequently, the aircraft returned to Marina
following the initial outbound path. To visualize the location and general timing of the wind reversal (Fig. 6b-c), 48-
hr back-trajectories from HYSPLIT were used. This contrasts with the 24-hr back trajectories used to confirm
southerly wind flow in Sect. 2.2. For the case studies, 48-hr periods were used to have a better understanding of air
mass history. This case of southerly wind is one where the sampled air mass was likely to have spent more time in the
coastal area just south of Marina as compared to traditional northerly flow, where there was presumed influence from
shipping emissions and possibly advected continental air.






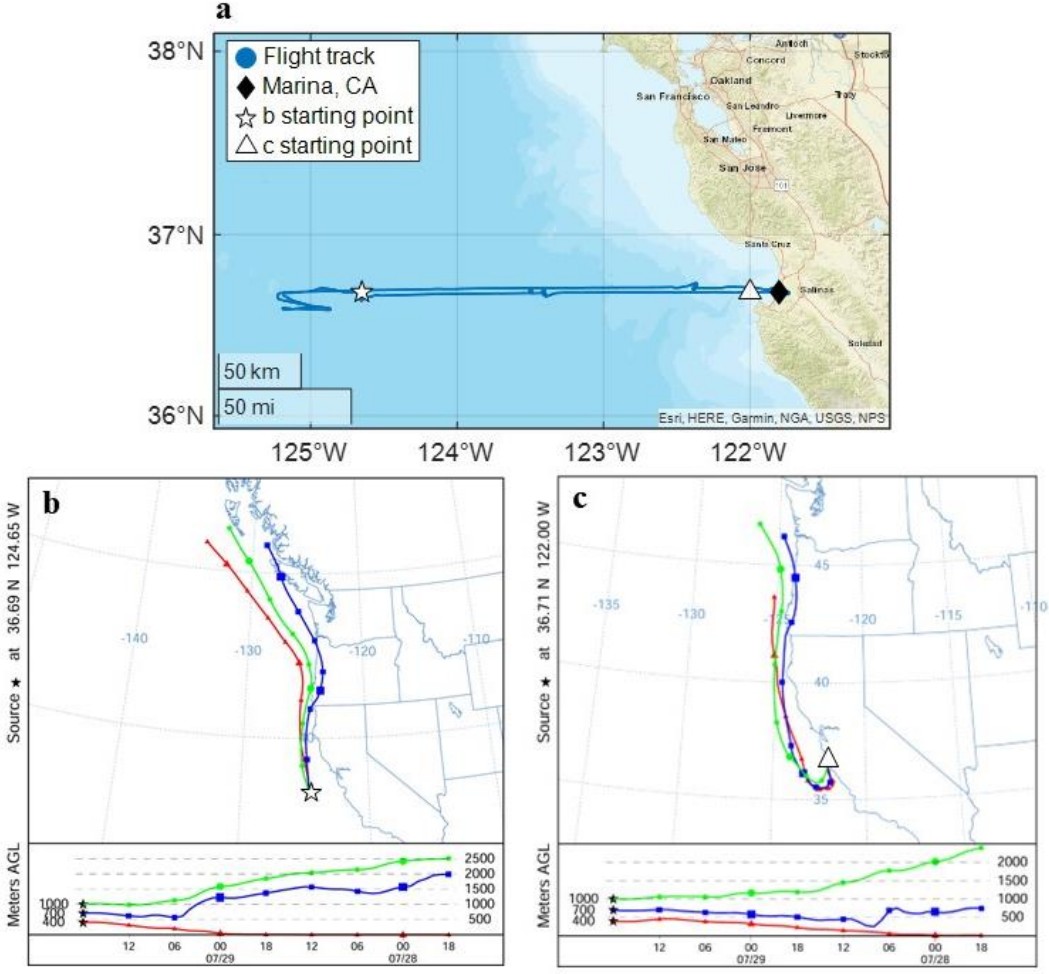

**Figure 6: (a) NiCE RF 16 (07/29/2013) flight track, with Marina represented by a solid black diamond, the starting point**
**of the HYSPLIT back-trajectory in panel (b) indicated by a white star, and the starting point of the HYSPLIT back-**
**trajectory in panel (c) indicated by a white triangle. (b) 48-hour back trajectory of a point (36.69° N, 124.65° W) along the**
**flight path outside of the southerly wind zone (HYSPLIT end time: 1800 UTC). (c) 48-hour back trajectory of a point (36.71°**
**N, 122.00° W) along the flight path at the beginning of the RF (HYSPLIT end time: 1700 UTC) where there was southerly**
**flow. Panels (b) and (c) detail back-trajectories for three different altitudes: 400, 700, and 1000 m.**
We investigated gradients from the coast to farther offshore including past the wind reversal for several
parameters, including $N_a$, $N_d$, and AMS total mass and mass fractions, both in the sub-cloud MBL (<525 m AGL, Fig.
7) and in the FT (>765 m AGL, Fig. S12), both altitudes of which were defined in Crosbie et al. (2016). There was a
general trend of decreasing number concentration, especially for $N_{a0.1-1\mu m}$, $N_{a>10nm}$, and $N_d$, from the coast to slightly
before the stacked legs at the far west point (1,245/189, 1,240/390, and 772/263 cm$^{-3}$, respectively, at ~1732/1830
UTC). There was a wide range of supermicron concentrations for the whole flight duration, however, generally, there
was a slight decrease of $N_{a>1\mu m}$ along the flight path going west as well, but it was not as pronounced as the other
variables (24/4 cm$^{-3}$).



The eastbound leg to Marina was an interesting situation as there was no longer southerly flow closer to the coast yet there was still a concentration increase for number and cloud drop concentrations but not up to the same maximum levels that were observed on the westbound portion of the flight, probably owing to the reduced influence from areas south of the sampling area ($N_{a0.1-1\mu m}$: 248/435, $N_{a>10nm}$: 454/752, $N_d$: 272/434, and $N_{a>1\mu m}$: 5/19 cm$^{-3}$, for eastbound/westbound legs at ~2000/2037 UTC). AMS mass concentrations dropped significantly in the outbound portion of the flight, from total mass as high as 10.16 µg m$^{-3}$ (~1730 UTC) to 1.55 µg m$^{-3}$ (~1745 UTC), the latter of which was approximately 10 km offshore. During that period, organic mass fraction decreased from 0.81 to 0.28 in favor of growing $SO_4^{2-}$ mass fraction from 0.11 to 0.50. On the inbound track, similar to $N_a$/$N_d$ results, there was not as much of an enhancement in total mass (max of 4.41 µg m$^{-3}$ at ~2040 UTC) and the chemical profile revealed more comparable levels of $SO_4^{2-}$ and organic mass fractions (0.39 and 0.52, respectively, at ~2040 UTC) in contrast to the outbound track that showed higher organic mass fraction right by the coast.

The results suggest that the enhanced residence time of air masses (due to the wind reversal) in an area with presumed influence from shipping emissions and continental pollution yielded an offshore gradient in $N_a$, $N_d$, and aerosol composition. Also, the results help show that this general coastal zone area in the location of the wind reversal is enhanced with fine pollution, which generally will affect aerosol and cloud characteristics if air masses spend prolonged time in it during southerly flow conditions.

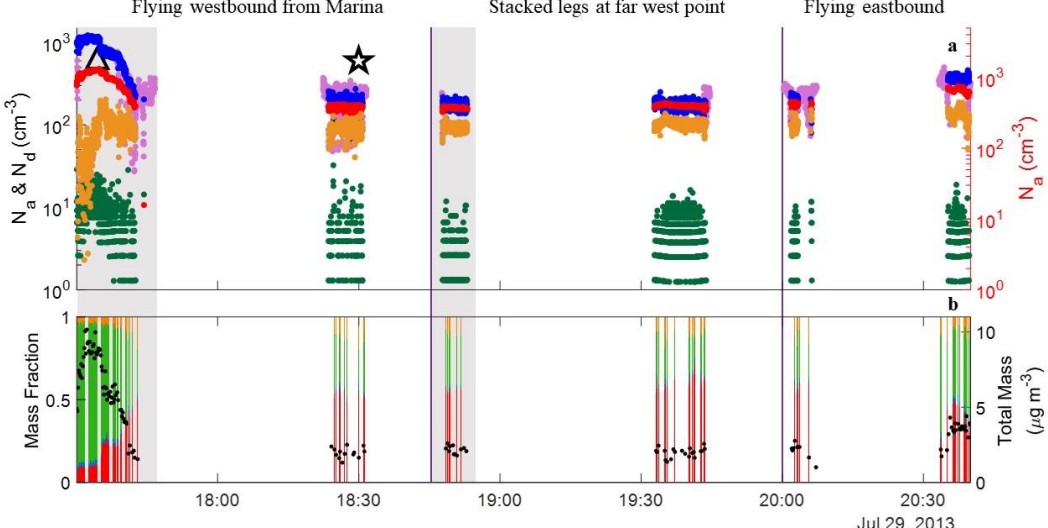

**Figure 7: Data from NiCE RF 16 in the MBL (<525 m). The grey shading indicates time periods with mostly southerly winds, and the purple lines across all graphs indicate flight zones (outbound track, stacked legs at farthest west point, and inbound track). (a) The colored points on the left-hand axis correspond to $N_{a0.1-1\mu m}$ (blue, PCASP$_{<1\mu m}$), $N_{a>1\mu m}$ (green, PCASP$_{>1\mu m}$), and $N_d$ (light purple, CASF). The colored points on the right-hand axis correspond to $N_{a>10nm}$ (red, CPC) and $N_{a10-100nm}$ (yellow, CPC 3010 – PCASP$_{<1\mu m}$). The triangle corresponds to the HYSPLIT back-trajectory end point seen in Fig. 6c, and the star corresponds to the HYSPLIT back-trajectory end point seen in Fig. 6b. (b) Stacked bar plot of AMS mass fractions of $SO_4^{2-}$ (red), $NO_3^-$ (blue), organics (green), and $NH_4^+$ (orange), overlaid with total mass concentration (µg m$^{-3}$; black).**

The trends in the FT are much more ambiguous than those in the MBL (Fig. S12). Similar to the MBL, there was a decrease in $N_{a0.1-1\mu m}$ and $N_{a>10nm}$ from the coast to near the stacked legs (2,467/395 and 2,820/689 cm$^{-3}$, respectively, at ~1726/1844 UTC), however there was no discernable trend for $N_{a>1\mu m}$. There were no apparent offshore trends for AMS total mass or speciated mass fractions. Additionally, on the eastbound flight leg, there was not a clear trend for any of the parameters. This suggests that the effects of the southerly winds were stronger in the MBL than the FT.



### 3.4.2 CSM Research Flight 6

CSM stands out among all of the examined campaigns owing to the strength and temporal persistence of wildfire plumes, which was also the main focus of the mission. Of the top 3% (n = 12) of the largest fires in CA in the historical record, four occurred in 2020 (circled in Fig. 3): the August Complex fire (16 August, Mendocino County), the SCU Lightning Fire Complex (18 August, Santa Clara County), the Creek fire (4 September, Madera County), and the LNU Lightning Complex fire (16 August, Hapa County) (Keeley and Syphard, 2021). These four fires were a mix of both merged (August Complex) and unmerged (LNU Lightning Complex) fires that burned over 417, 160, 153, and 146 kha, respectively, and burned for months after they were ignited.

CSM RF 6 (10 September 2020) included two major components (Fig. 8a): a spiral over Salinas (max altitude of 6,172 m at ~2000 UTC) and a spiral over Monterey Bay (max altitude of 4,822 m at ~ 2170 UTC). The entire region was heavily impacted by smoke during CSM RF 6 (Fig. 8b). Additionally, around 36.5° N, 125° W, there is an area not dominated by smoke, but rather, clouds, pointing to the likelihood of smoke-cloud interactions in the region on not just this day but other CSM days with similar smoky conditions. HYSPLIT back-trajectories for the two spirals for a 48-hr period were generated (Fig. 8c and 8d). For the spiral over Monterey Bay (Fig. 8c), the lowest altitude trajectory (trajectory beginning at 400 m) is mostly northwesterly, the second lowest altitude (trajectory beginning at 1400 m) is primarily southerly, and the highest altitude (trajectory beginning at 2400 m) is approximately northeasterly. The highest altitude back-trajectory passes over the LNU Lightning Complex fire (red oval; circled in Fig. 3). For the spiral over Salinas (Fig. 8d), all three altitude levels (400, 800, and 1200 m AGL) reveal southerly trajectory paths, and the air masses from the second-highest altitude back-trajectory possibly had some influence from the SCU Lightning Fire Complex (purple oval) and the August Complex Fire (green oval) due to offshore and northerly flow in the preceding 36-hr (Fig. 3).

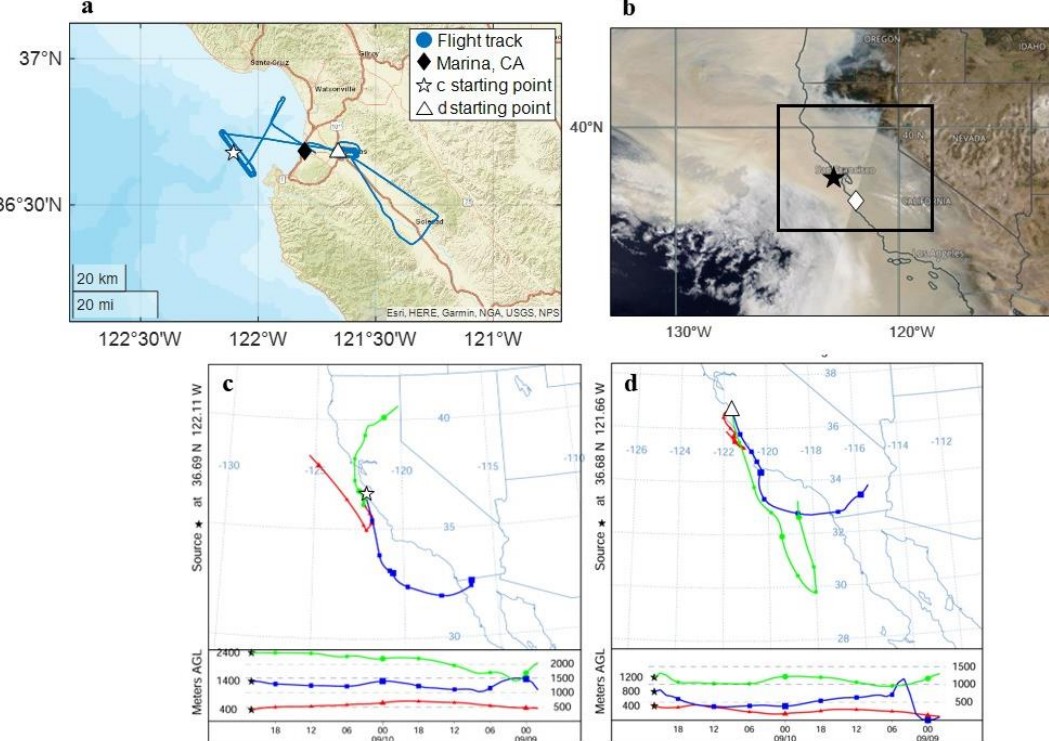

**Figure 8: (a) CSM RF 6 (09/10/2020) flight track, with Marina, CA represented by a solid black diamond, the starting point of the HYSPLIT back-trajectory in panel (c) indicated by a white star, and the starting point of the HYSPLIT back-trajectory in panel (d) indicated by a white triangle. (b) NASA Worldview image, with Marina, CA represented by a white**




**diamond, and Pt. Reyes denoted by a black star. (c) 48-hour back trajectory of a point (36.69° N, 122.11° W) along the flight**
**path during the sounding over Monterey Bay (HYSPLIT end time: 2100 UTC) at three different altitudes: 400, 1400, and**
**2400 m. (b) 48-hour back trajectory of a point (36.68° N, 121.66° W) along the flight path during the sounding over Salinas**
**(HYSPLIT end time: 1900 UTC) at three different altitudes: 400, 800, and 1200 m. (c) and (d) utilized different altitudes**
**for the back-trajectories to reflect the different maximum altitudes of the two major soundings of the flight.**

The vertical profiles of temperature, wind speed, and wind direction are provided in Fig. S13 for context.
Notably, the vertical region with southerly flow was thicker over the ocean (approximately 370 – 3700 m) versus over
land (540 – 2900 m). $N_a$ for different size ranges and $N_{a3}$:$N_{a10}$ are shown separately for land and over the ocean (Fig.
9). There was more variability in $N_{a>10nm}$ (Fig. 9a) over the ocean, with a general decrease in concentration with
increase in altitude for both data over land and ocean, followed by increasing $N_{a>10nm}$ above of the region of primarily
southerly flow (non-shaded points). There was not much change in $N_{a>1\mu m}$ (medians = 1 – 3 cm⁻³; range = 0 – 6 cm⁻³;
Fig. 9c) until >2.5 km, where concentration increases over land (medians = 5 – 97 cm⁻³; range = 0 – 297 cm⁻³) where
there is primarily northerly flow, likely from sampling smoke plumes. Over the ocean, low supermicron particle
concentrations are observed (≤ 7 cm⁻³). These results show that during extensive smoky periods, the flow regime does
not matter in cases like RF6 due to smoke generally being all across the region. Furthermore, the results show that
supermicron particle concentrations are certainly enhanced in smoke plumes, as has been observed before in the study
region (Mardi et al., 2018) but not to this pronounced extent, especially at high altitudes over land.
The $N_{a3}$:$N_{a10}$ ratio (Fig. 9d) was generally consistent over land across all vertical levels, with a good number
of outliers in the region of primarily southerly flow. The medians of the ratios over the ocean were usually lower than
the medians over land until 3.5 km. There was no discernable difference in the $N_{a3}$:$N_{a10}$ ratio over land between
southerly and northerly flow (medians approximately 1.35 until >5.5 km) or over the ocean (medians for both flow
regimes approximately 1.20, with a slight bump to 1.26 and 2.14 between 3.5 and 4.5 km).

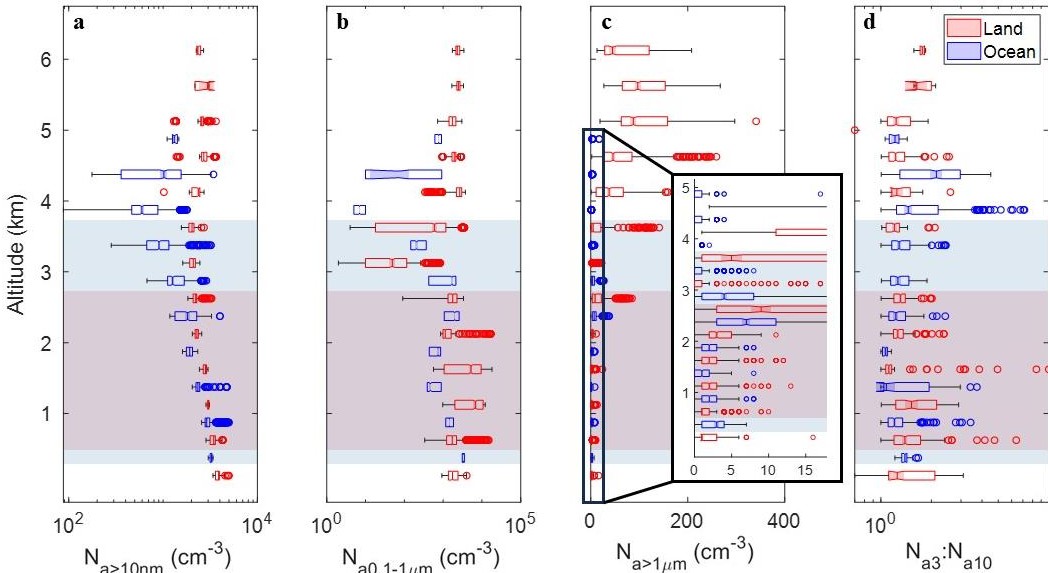

**Figure 9: CSM RF 6 box plot vertical profiles of (a) $N_{a>10nm}$ (cm⁻³), (b) $N_{a0.1-1\mu m}$ (cm⁻³; PCASP$_{<1\mu m}$), (c) $N_{a>1\mu m}$ (cm⁻³;**
**PCASP$_{>1\mu m}$), and (d) $N_{a3}$:$N_{a10}$. Data are shown every 500 m over land (red) and ocean (blue) above the MBL, which is the**
**maximum altitude of the first bins for all the panels. Panel (c) has an additional focus on altitudes ≤5 km ($N_{a>1\mu m}$ ≤ 18 cm⁻³**
**³). The red and blue shading indicates altitudes over the land and ocean, respectively, with southerly winds.**
Complementary data from NAAPS and COAMPS are shown in Fig. 10 for this case flight. COAMPS and
NAAPS (Fig. 10a and 10b, respectively) both show southerly winds generally in the outlined study domain, which is



consistent with observational data showing southerly winds close to Marina. NAAPS shows stronger southerly winds
over land near Marina compared to over Monterey Bay whereas there was not much of a difference in wind speed
between the two spiral soundings from the Twin Otter (Fig. S13). COAMPS better simulates southerly flow along the
coastline, whereas the spatial resolution of NAAPS is probably a reason for it not being able to capture southerly flow
in the grid spaces closest to the coast especially just south of Marina – instead there is weak northerly flow.
A notable difference between NAAPS and COAMPS when it comes to modeling smoke (Fig. 10c and 10d,
respectively) is that NAAPS better represents smoke over the ocean and more closely matches the visible satellite
imagery from Fig. 8b. COAMPS does not capture smoke over the ocean away from the coastline. We do not focus on
comparing absolute mass concentrations of smoke as it is difficult to know the ground truth value from the aircraft
observations and also because of the different ways and size classifications for smoke in the two models. Generally,
though, NAAPS and COAMPS match in the general areas identified as having smoke and areas of high concentrations
match one another. Looking at Fig. 10e, NAAPS shows high concentrations of sea salt offshore west of 130° W.
However, near the flight area and within our region of focus, sea salt concentrations are less than 5 µg m$^{-3}$. NAAPS
ABF (Fig. 10f) mirrors the areas with areas of high sea salt in Fig. 10e, but similar to model results from Sect. 3.2.2.4,
there are areas of higher ABF concentrations (2 – 3 µg m$^{-3}$) near the ports of Los Angeles and Long Beach (34° N,
118° W) as well as up north near San Francisco and San Jose (38° N, 122° W). NAAPS dust (Fig. 10g) and coarse
mass (Fig. 10h) also resemble the areas with high sea salt, with coarse mass concentrations exceeding 10 µg m$^{-3}$ near
both Marina, CA and Pt. Reyes.



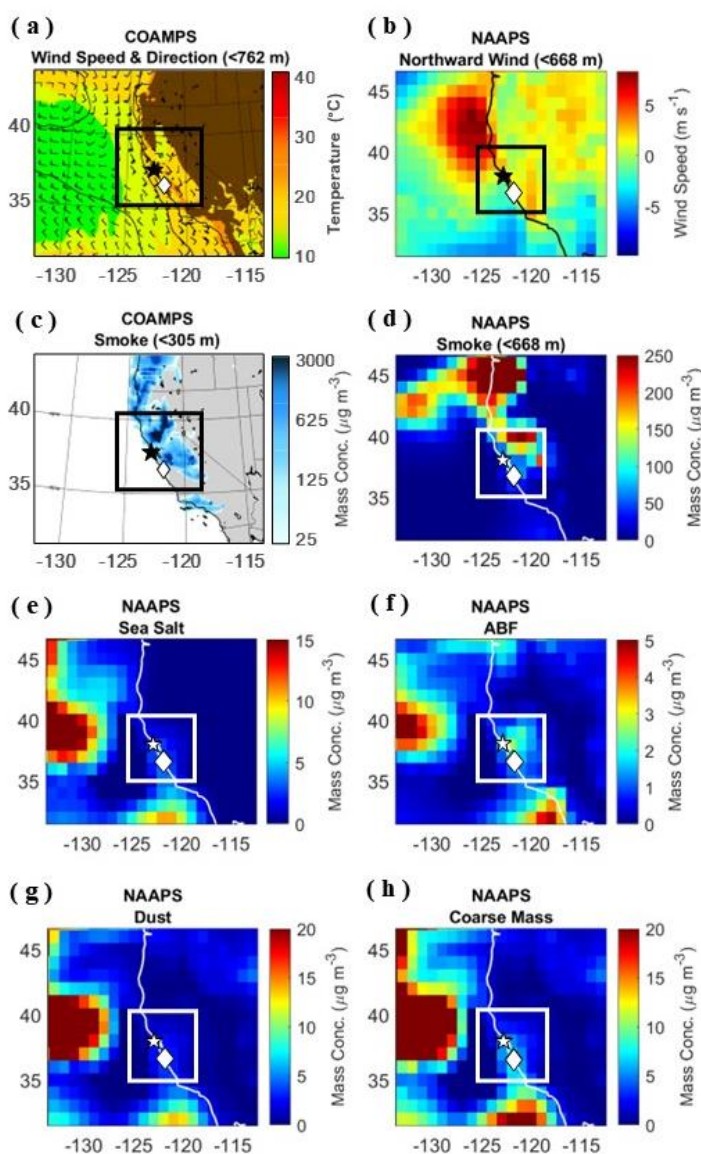

**Figure 10: COAMPS/NAAPS images are for 2100 UTC. (a) Wind speed and direction up to 762 m derived from COAMPS. The colors indicate surface temperature (°C). (b) Wind speed of northward wind up to 668 m derived from NAAPS. (c) Smoke concentration (µg m⁻³) up to 305 m derived from COAMPS. (d) Smoke, (e) sea salt, (f) ABF, (g) dust, and (h) coarse mass concentrations (µg m⁻³) up to 668 m derived from NAAPS. The white diamond indicates Marina, CA, the white star indicates Pt. Reyes, and the black & white boxes indicate our zone of interest.**

## 4    Conclusions

In this study, we utilized multiple types of data, including a large repository of NPS Twin Otter data, to compare coastal aerosol and cloud characteristics near central CA for northerly and southerly wind regimes in the lower troposphere. Juliano et al. (2019a) had previously called for future studies to utilize in situ observations to support their investigation into cloud properties using satellite observations. Our study is the first to investigate aerosol



and cloud droplet number concentrations through in situ aircraft data in addition to CW composition, and intercompare
those results with satellite data, as well as models and surface station data.
Our first hypothesis is proven correct in that more fine aerosol pollution is present off the CA coast during
southerly flow due to influence from shipping exhaust and continental emissions including from major cities like Los
Angeles. Submicron aerosol pollution is found to be higher during southerly flow days (particularly during E-PEACE),
with respect to both $N_a$ ($N_{a>10nm}$, $N_{a10-100nm}$, $N_{a0.1-1\mu m}$) and concentrations of shipping and continental tracer species in
surface data ($SO_4^{2-}$, $NO_3^-$, OC, V, Ni, and EC) and CW samples (nss-$SO_4^{2-}$, $NO_3^-$, $NH_4^+$, V and oxalate). Cloud water
is shown to be more acidic during southerly flow along with more $Cl^-$ depletion based on lower $Cl^-$:$Na^+$ ratios. A
secondary hypothesis was that increased influence from shipping and/or continental emissions would lead to enhanced
$N_d$ and COT and lower $r_e$ (at fixed LWP) due to the Twomey effect (Twomey, 1974). Both the airborne in situ data
and satellite retrievals show increased $N_d$ on southerly days. The satellite retrieval data also reveal higher COT and
lower $r_e$ during southerly flow The increase in $N_d$ and decrease in $r_e$ associated with the northerly to southerly reversal
matches results of a previous study in the region (Juliano et al., 2019a). The analysis of CSM RF 6 reveals that during
heavy biomass burning periods with prevailing smoke, there is relatively no difference in aerosol or cloud properties
associated with changes in flow regime.
A limitation in this type of study to address in the future is the difficulty of obtaining detailed in situ data
during southerly wind conditions. As noted already, wind reversals along coasts extend to a number of other global
regions (e.g., South America, southern Africa, Australia) and thus it is recommended to continue building more
statistics to better understand changes in aerosol and cloud properties as a function of wind direction along coastal
regions. Intercomparisons with models, as partly done here, can aid with determining if model resolution should
improve to better simulate these events. Generally speaking, the prevalence of fine aerosol on southerly flow days and
associated changes in cloud microphysical properties are important findings with implications for weather, health,
coastal ecology, and aviation.

**Data availability**

Airborne data used in this work can be accessed at https://doi.org/10.6084/m9.figshare.5099983.v11 (Sorooshian et
al., 2017). Bouy data from the NOAA's NDBC can be accessed at https://www.ndbc.noaa.gov/. The archived data
from GOES-West Full Disk Cloud Product (GOES-15) can be accessed at https://satcorps.larc.nasa.gov/. The archived
surface weather plots from NOAA's WPC can be accessed at
https://www.wpc.ncep.noaa.gov/archives/web_pages/sfc/sfc_archive.php. The surface data from IMPROVE can be
accessed at http://views.cira.colostate.edu/fed/. The MODIS-Aqua data can be accessed through NASA Giovanni at
https://giovanni.gsfc.nasa.gov/giovanni/. The FIRMS data can be accessed at https://earthdata.nasa.gov/firms.

**Author contributions**

AW and PX aided with access and interpretation of COAMPS and NAAPS data, respectively. KZ and GB conducted
the data analysis. KZ and AS conducted data interpretation. KZ and AS prepared the manuscript. All authors edited
the manuscript.

**Competing interests**

At least one of the (co-)authors is a member of the editorial board of Atmospheric Chemistry and Physics.

**Disclaimer**

Publisher's note: Copernicus Publications remains neutral with regard to jurisdictional claims in published maps and
institutional affiliations.




## Acknowledgements

The authors acknowledge NPS staff for successfully conducting Twin Otter flights and all others who were involved
in the airborne campaigns. We thank Ewan Crosbie for useful discussions about this work.

## Financial support

This work was funded by Office of Naval Research grant N00014-21-1-2115.

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
