# Peer review of "Differences in aerosol and cloud properties along the central California coast when winds change from northerly to southerly"

_EGUsphere, 2024_

## Referee Comment (RC2)

**Review of "Differences in aerosol and cloud properties along the central California coast when winds change from northerly to southerly" – Zeider et al. (2024)**

The authors present an analysis of aerosol-cloud interactions in marine stratocumulus off the western coast of the United States based on a combination of field campaign data, satellite data, buoy data, and modeling. They extend the work of Julian et al. (2019, JAS), who focused on just 3 cases, showing that many of the results presented in the earlier work are robust and extend to more cases.

I feel that this research is worthy of publication and will be of interest to a broad portion of the community after the comments listed below are addressed.

If the authors have any questions, please do not hesitate to contact me.

Zachary J. Lebo

Major Comments

1) **Modeling:** The motivation for the modeling portion of this study is lacking in my opinion. In particular, if the simulations are intended to be used to study aerosol-cloud interactions, then I would argue that the grid spacings used in this work are way too coarse. So up front the motivation and what you expect to learn from these simulations needs to be better presented, otherwise the results gleaned from the model simulations are not as meaningful or impactful, let alone even accurate (again if the goal is aerosol-cloud interactions, which occur on much finer spatial scales).

2) **Synoptic conditions:** I am concerned that by not providing some synoptic analysis, it is hard to understand why some of the results exist and why they differ from prior studies. Is it partially a sampling issue? If just 2 days have southerly flow but 10 do not for a given field campaign, the data for the 10 days should be more robust than for the 2 days? In Juliano et al. (2019, BAMS), this was addressed by selecting a sample of non-CTD days that was the same size as the CTD sample. If this was not done in the current study, it might be good to ensure that the datasets are consistent in this manner.

Minor Comments

1) Figure 1: I realize that the flights for the different campaigns span different areas, but it is quite difficult to compare the regions that were sampled owing to the different axes used in each panel. I would suggest using a single set of axes for each plot for consistency. It would also be beneficial to know what percentage of each RF was i) over the open water as well as ii) in cloud. I say this because those are the samples that are of interest to this study, and while the study does expand on the number of cases used in prior work, it is hard to tell exactly how much the dataset is expanded. Many of the flights appear to be largely over land, which would diminish the size of the relevant data.

2) Line 205: This line makes references to thresholding data for when the aircraft was over the ocean, which further emphasizes my point made above about quantifying that for the reader.

3) Lines 232-233: How was this done exactly? I ask because assessing wind speed and direction from cloud motion works if the clouds follow the motion of the air, i.e., act as a tracer. In many cases, they do not act as a tracer of air motions, e.g., mountain wave clouds and the stratocumulus decks that are the focus of this study. So I think more details are needed here to provide confidence that the cloud motion extracted from the geostationary satellite data is in fact close to the wind speed/direction.

4) Line 240: This location should be denoted on a map. Why was this location used?

5) Line 260: I would suggest calling this the "meridional" or "north-south" wind and then highlight that north is positive.

6) Lines 263-265: Is there a sounding near the coast that you could use to show that this is a reasonable assumption?

7) Lines 305-306: This is probably just my naiveness with these data, but why only every 3rd day?

8) Lines 345-346: Could you plot the wind profiles to perhaps show that at the surface, the southerly flow is larger, but with height that flow weakens and leads to a mean decrease in wind speeds?

9) Table 3 (and elsewhere): This may just be a personal thing but the use of "/" to separate values is sometimes confusing. I had to look back in a few instances to make sure that it was showing two different values and not intended to be a ratio. I might suggest being more explicit with the values and not use "/" even if the text becomes a bit longer.

10) Figure 2: I do not see a difference in the sign of the wind in the top left two panels. Both northerly and southerly flow have positive values? Is this just a mistake in the plotting?

11) Line 361: The use of "Southerly–Northerly Days" is confusing. I am not sure if this is the transition day? A difference between southerly days and northerly days (I think that is what it is).

12) Lines 363-364: I am confused...it says above in the paper that by definition northerly is positive, so how could the northerly winds be less negative?

13) Lines 372-374: This sentence is very confusing; suggest rewording to better convey the meaning.

14) Figure 2 (again): Suggest using a colormap with white in the middle to make it easier to see positive vs negative values.

15) Lines 385-387: Fair enough that in some cases, the fires were not right near the sampling site, but smoke can be advected for long distances, so how can you be certain that these campaigns were not affected by the more distant wildfires?

16) Line 387: This is the first mention of "Marina". Having taken part in EPEACE, I know what you are talking about, but the general audience may not. I would be more clear with where this is.

17) Figure 4: Suggest adding a y axis label to say what is shown, not just the units.

18) Line 523: Define "reasonable"?

19) Lines 561-562: Suggest adding a sentence or two at least speculating on why supermicron aerosol could change (i.e., put forth a hypothesis or two and then proceed with the analysis to confirm/deny).

20) Lines 599-606: This was something that we struggled with in Tim's papers was whether the aerosol were influencing the clouds or the meteorology was influencing the clouds? Any thoughts?

21) Figure 5: It might be good to label the y axes and not just the top of the plots.

22) Lines 779-791: I feel like the modeling here distracts from the main theme. This also loops back to major comment number 1.
23) Lines 804-806: I think Juliano et al. (2019, JAS) did use all of these data sources, but maybe not in the same proportion as in the current study?
24) Figure S10: The wind speeds in the MACAWS case are quite strong, approaching 25 m/s based on Fig. S10. Any idea what was going on to cause such winds?

---

## Author Comment (AC1)

We thank the two reviewers for their helpful comments. We have provided our responses to comments below in blue.

REFEREE 1

This study utilizes ground-based and airborne *in situ* measurements and two different numerical models (one a global, coarse-gridded aerosol model with prescribed dynamics, the other a mesoscale dynamical model with interactive aerosols) to more deeply examine the assertion that coastally-trapped disturbances (CTDs)/southerly surges result in enhancements of aerosol and cloud droplet loading in near-coastal regions. Satellite measurements (the primary source of observational data used in previous studies on this topic) are also utilized for broader context. The airborne measurements are from a number of local summertime field experiments spanning about a decade, but the rarity of CTDs and the fact that none of the campaigns considered were explicitly focused on sampling them means there are relatively few cases analyzed (17 out of 114 research flights across 6 field campaigns). This is even more so the case for the ground-based measurements due to the fact that sampling was only performed every 3 days.

The authors find a rather variable relationship between CTD occurrence and aerosol loading compared to Juliano et al. (2019), who examined satellite retrievals and reanalysis data and specifically selected "strong" CTD cases. Perhaps this should not come as a surprise given that the region sampled by the aircraft (typically in the vicinity of Monterey Bay) is shown by Juliano et al. (2019) to have a more muted microphysical response than locales farther offshore and to the north (see their Fig. 5). Despite the more variable CTD/non-CTD relationship, the authors do generally find enhanced aerosol loading (especially in the accumulation mode) and cloud drop number concentration, with further supporting evidence from cloud water composition measurements that show enhanced concentrations of anthropogenic combustion tracers (e.g., NO3-, V, elemental carbon). A few case studies seek to explore the observed relationships in greater detail using HYSPLIT back trajectories and aircraft measurements. I did not find these case studies to add much to the discussion, and in fact they may raise more questions for readers (see Major Points below).

Overall, the study is worthy of publication in ACP after the authors address the comments below. I look forward to seeing the next iteration of the paper.

Mikael Witte

Response: We thank Dr. Witte for the positive reply.

**MAJOR POINTS**

**Case studies**:

The NiCE case study appears to have been a relatively weak (or perhaps incipient?) CTD based on the back trajectories shown in Fig. 6. While it is obvious that Na is considerably greater during the westbound segment of the flight, it's hard to buy the argument that this was due to

airmasses spending a significant amount of time in a busy shipping lane. The back trajectory in Fig 6c shows the airmass spending a decent amount of time offshore of Big Sur, but to my knowledge this is not exactly a major shipping lane. I also find it hard to believe that pollution from LA/Long Beach is making it this far north. So what other sources of pollution could be impacting this area? My guess is that the NAM may not be capturing smaller-scale/intermittent offshore flow events along the Oregon and Northern California coasts, but I don't see an easy way to prove this.

Response: Thank you for this comment. The reviewer comments in general point to the need in our case to make it more clear that there is a lot of complexity in what is going on with the measured values related to aerosol properties and that it is too simplistic to assume shipping emissions can be a major explaining factor for this case study. There can be influence from aloft for instance representing continental emissions that may potentially be significant in affecting the results. We find this case study to be relevant to the discussion even though it doesn't provide many answers, but instead it helps motivate more focus on the general wind/dynamic patterns in this study region. Philosophically, we view these case studies as being seeds in a way to promote more work to help us understand the controlling factors over aerosol and cloud properties off the California coast.

We added text in a number of places in the paper to highlight these complexities:

Sect. 3.4: "These case studies help emphasize the complexity of flow patterns in the region that influence the ability of aerosols from different sources to arrive at the boundary layer in the study region. The observed changes in aerosol and cloud properties between northerly and southerly days are likely not due to an instant switch in flow direction but rather there is critical nuance in the timing, strength, and duration of the wind reversal, along with likely influence from free tropospheric aerosol which can be sourced from various continental areas across California and even farther away (Dadashazar et al., 2019)."

Sect. 3.4.1: "The results suggest that the enhanced residence time of air masses (due to the wind reversal) in an area with presumed influence from shipping emissions (see Fig. 9 in Coggon et al., 2012) and continental pollution yielded an offshore gradient in $N_a$, $N_d$, and aerosol composition. Also, the results help show that this general coastal zone area in the location of the wind reversal is enhanced with fine pollution, which generally will affect aerosol and cloud characteristics if air masses spend prolonged time in it during southerly flow conditions. This all being said, it is hard to unambiguously attribute the aerosol and cloud changes to emissions from a particular area and source due to the complex flow nature in both the horizontal and vertical directions during the wind reversal period. This case study helps motivate continued research studying these events."

Conclusions: "We caution that there is considerable complexity in flow patterns both horizontally and vertically when northerly winds change to southerly winds and this warrants more research to study for instance how influential free tropospheric air is for the boundary layer aerosol changes occurring on southerly flow days."

We also have added boundary layer flow patterns to the supplement to more clearly capture the differences in flow regime between northerly and southerly flow days, along with the following accompanying text:

"For context, boundary layer flow patterns from NAVGEM are provided in Fig. S13 for all southerly and northerly days (Fig. S14 and S15 provide flow maps for each individual campaign). The average southerly flow pattern (Fig. S13a) captures generally weaker flow, particularly near Marina, CA, where a slight reversal can be observed. When looking at the flow maps for each campaign (Fig. S14 and S15), only BOAS and FASE captured a small wind reversal by Marina, CA during southerly flow days. Both MACAWS and CSM had a circulatory-pattern north of Marina, CA, near Pt. Reyes, and southerly flow is more clearly observed during the CSM campaign along the coast."

[Figure]

**Figure S13: NAVGEM boundary layer flow patterns for (a) all southerly flow days and (b) all northerly flow days included in this study. These results are for 1800 UTC for all days of the campaign months in Table 1 as was done for Figs. S4-S10; the lowest level of the model was used representing the lowest ~50 m. The airport in Marina, California is denoted by a white diamond and Pt. Reyes is indicated with a black star.**

[Figure]

**Figure S14: NAVGEM boundary layer flow patterns for (left) southerly and (right) northerly flow days during (a-b) E-PEACE, (c-d) NiCE, and (e-f) BOAS. These results are for 1800 UTC for all days of these respective campaign months in Table 1; the lowest level of the model was used representing the lowest ~50 m. The airport in Marina, California is denoted by a white diamond and Pt. Reyes is indicated with a black star.**

[Figure]

**Figure S15: NAVGEM boundary layer flow patterns for (left) southerly and (right) northerly flow days during (a-b) E-PEACE, (c-d) NiCE, and (e-f) BOAS. These results are for 1800 UTC for all days of these respective campaign months in Table 1; the lowest level of the model was used representing the lowest ~50 m. The airport in Marina, California is denoted by a white diamond and Pt. Reyes is indicated with a black star.**

The CSM case, on the other hand, *does* exhibit substantial southerly flow both within and above the boundary layer, but from Fig. 3, the smoke source is apparently to the north and east! So are the extremely high smoke/aerosol concentrations a consequence of advection in southerly flow or more of a "ping-ponging" effect in which smoke is advected over the area with the (climatological) northerlies at an earlier time and then essentially held in place/re-circulated over your sampling region with the reversal to southerly winds?

Response: Yes, exactly – we also believe there is a ping-ponging effect. We have added in boundary layer flow pattern maps to the Supplement, including a distinction between southerly and northerly flow days for CSM. Please see added text with respect to this comment:

"As illustrated by the composite boundary layer flow pattern in Fig. S15e-f, smoke along the coast during southerly flow periods was re-circulated northwest of Marina, CA nearby the flight path (which was not observed for the northerly composite flow pattern), which could have also influenced the elevated aerosol concentrations during this flight."

The fact that these two cases are rather complicated points to a much more nuanced picture regarding how aerosols arrive at the marine boundary layer than is given by Juliano et al. (2019), and that was the main point I came away with from the case studies (and really, the paper as a whole). I think it would be helpful if your summary explicitly recognized this fact – it's not like the wind flips to southerly and there's an instantaneous and unambiguous increase in Na/Nd. The timing of a reversal, its strength and duration (which determine the southward fetch and impacts the probability of, e.g., SoCal-sourced particulate), and the availability of aerosol (from smoke, shipping, etc.) are interconnected factors that determine how much "extra" aerosol makes it into the boundary layer, and this was more or less glossed over in Juliano's BAMS paper. If one of the points of your paper is to motivate further research on this topic, acknowledging how much the details matter in determining the unfolding of individual events would make for a stronger argument.

Response: We appreciate this comment and agree that we should explicitly point out the many interactive factors that may affect how aerosols arrive in the boundary layer of the study region.

We added this text:

"These case studies help emphasize the complexity of flow patterns in the region that influence the ability of aerosols from different sources to arrive at the boundary layer in the study region. The observed changes in aerosol and cloud properties between northerly and southerly days are likely not due to an instant switch in flow direction but rather there is critical nuance in the timing, strength, and duration of the wind reversal, along with likely influence from free tropospheric aerosol which can be sourced from various continental areas across California and even farther away (Dadashazar et al., 2019)."

**SoCal aerosol sources**: I don't see any clear evidence for an aerosol source from Southern California. Unless you can point to something(s) in the observations that would support this idea, I suggest you significantly scale back your discussion of this point and remove it from the conclusions. It seems just as likely that "local" emissions from the Bay Area (or more broadly, coastal Northern California) could be the source of the combustion tracers in the NiCE case study, and it's obvious that wildfire smoke dominates the CSM case; perhaps you can find a clearer example of a back trajectory that passes over SoCal from another case?

Response: We have tried to soften anything close to a confident claim that southern CA sources are leading unambiguously to high aerosol levels in the study region on southerly flow days.

We modified the text in the Conclusions to omit usage of southern California cities:

"Our first hypothesis is proven correct in that more fine aerosol pollution is present off the CA coast during southerly flow due to likely influence from shipping exhaust and continental emissions. We caution that there is considerable complexity in flow patterns both horizontally and vertically when northerly winds change to southerly winds and this warrants more research to study for instance how influential free tropospheric air is for the boundary layer aerosol changes occurring on southerly flow days."

We have also added in an example of a back-trajectory from MACAWS that is a clearer example of a back-trajectory that passes over SoCal into the supplement, which is provided below:

[Figure]

**Figure S17: 72-hour HYSPLIT back-trajectory ending on 30 July 2018 by Marina, CA (indicated by the black star).**

We added some text in the paper to mention this supplemental figure:

"This suggests that while there were elevated levels of anthropogenic emissions in this area regardless of the flow regime, there were increased concentrations during southerly flow days according to NAAPS. An example HYSPLIT back-trajectory for a southerly flow day (Fig. S17) shows air masses with likely influence from as far south as southern California and the U.S.-Mexico border."

**COAMPS**: I only see COAMPS data in one figure – did you analyze COAMPS output for any other days than the CSM case study? Do you have a general sense for whether NAAPS (as a proxy for "coarse-gridded models in general") can produce proper southerlies vs. weaker northerlies? Given the very minor role COAMPS simulations play in the current version of the paper, I question whether the conclusions drawn from a single case study add much to the manuscript.

Response: We appreciate this comment. We have moved the COAMPS analysis to the Supplement as it slightly distracts from the main themes of the paper. Additionally, we have added this text with respect to our general sense for whether coarse-gridded models can produce proper southerlies:

"Based on the NAAPS evaluation, while coarse-gridded models can capture differences in wind direction and aerosol concentration between southerly and northerly flow days, they are not fully able to reproduce southerly flow."

**MINOR POINTS**

(all refer to specific line numbers XX-YY, abbreviated LXX-YY)

L174: Is there any sensitivity of the results to your chosen LWC threshold?

Response: This is an excellent question and something we have worked a lot on internally in terms of seeking the best cloud-screening criteria. In fact, this can be given its own study in a way to explore sensitivities of aerosol and cloud variable values (and their relationships) to different criteria. In our case, we didn't see any noticeable difference in results by slightly altering the LWC threshold and don't get into such details in the paper as it would probably be viewed as distracting.

L760-761: re: the number of outliers in southerly flow in Fig. 9d – do you have sufficient sampling to say with certainty whether these are "outliers" or are you undersampling the distribution of Na3:Na10?

Response: With respect to the following breakdown of each box plot over the ocean and over land per altitude bin, the majority of the box plots had less than 5% of points that were displayed as outliers in Fig. 9d. There were some cases where there was greater than 5% of total points for an altitude bin being displayed as outliers (mostly over land), however we do not consider these cases to indicate an overall undersampling of $N_{a3}$:$N_{a10}$ for this case study. Therefore, no changes have been made to the manuscript.

| Ocean | | | Land | |
| Points/Outliers | % Outliers | Altitude (km) | % Outliers | Points/Outliers |
| --- | --- | --- | --- | --- |
| 65/2 | 3.08 | 0.5 | 0.00 | 144/0 |
| 340/16 | 4.71 | 1 | 2.60 | 269/7 |
| 102/2 | 1.96 | 1.5 | 0.00 | 121/0 |
| 34/0 | 0.00 | 2 | 16.00 | 75/12 |
| 104/3 | 2.88 | 2.5 | 8.01 | 137/11 |
| 109/0 | 0.00 | 3 | 2.50 | 200/5 |
| 225/7 | 3.11 | 3.5 | NA | NA |
| 375/23 | 6.13 | 4 | 5.45 | 55/3 |
| 253/0 | 0.00 | 4.5 | 4.55 | 22/1 |
| 14/0 | 0.00 | 5 | 2.58 | 194/5 |
| NA | NA | 5.5 | 0.81 | 124/1 |
| NA | NA | 6 | 0.00 | 6/0 |
| NA | NA | 6.5 | 0.00 | 18/0 |

**TYPOGRAPHICAL POINTS**

L90: "an important inventory…*is* leveraged…"

Response: Change made.

L91: "increased statistics" – unclear wording. I suggest "improved sampling" or perhaps "increased sampling density." I noticed this wording elsewhere as well – doesn't make sense to increase statistics themselves. What we really want is more data points.

Response: Change made in various places that "statistics" is mentioned including the line mentioned in the comment.

L118: "spaceborne" vs "space-borne"

Response: Change made.

L174: why not ">" instead of "needing to exceed"? similarly, use "<" instead of text "less than"

Response: Change made.

L179-180: Suggest rewording to "The mode wind direction was calculated for…"

Response: Change made.

L752: "by increase $N_{a>10nm}$ above the region…" (remove "of" from quoted phrase)

Response: Change made.

L783: suggest "different methods" vs "different ways"

Response: Change made.

L831: "Buoy" vs "Bouy"

Response: Change made.

In closing, we thank Dr. Witte for these excellent comments, and we probably agree mutually that overall, it is an important area of research to continue doing to better understand the complex nature of the flow behavior and influential aerosol sources during southerly flow events.

REFEREE 2

Review of "Differences in aerosol and cloud properties along the central California coast when winds change from northerly to southerly" – Zeider et al. (2024) The authors present an analysis of aerosol-cloud interactions in marine stratocumulus off the western coast of the United States based on a combination of field campaign data, satellite data, buoy data, and modeling. They extend the work of Julian et al. (2019, JAS), who focused on just 3 cases, showing that many of the results presented in the earlier work are robust and extend to more cases. I feel that this research is worthy of publication and will be of interest to a broad portion of the community after the comments listed below are addressed. If the authors have any questions, please do not hesitate to contact me.

Zachary J. Lebo

Response: We thank Dr. Lebo for the positive reply.

Major Comments

1) Modeling: The motivation for the modeling portion of this study is lacking in my opinion. In particular, if the simulations are intended to be used to study aerosol-cloud interactions, then I would argue that the grid spacings used in this work are way too coarse. So up front the motivation and what you expect to learn from these simulations needs to be better presented, otherwise the results gleaned from the model simulations are not as meaningful or impactful, let alone even accurate (again if the goal is aerosol-cloud interactions, which occur on much finer spatial scales).

Response: Thank you for this comment. We have clarified the motivation and our expectations from the modeling, which is as follows:

"The motivation for the usage of these models is two-fold. The NAAPS-RA has a coarse horizontal resolution; however, it provides large-scale aerosol conditions with observational constraints on the model fields (i.e., incorporates satellite retrieved aerosol optical depth). It is important to have this relatively accurate large-scale aerosol background information for regional aerosol-cloud interaction research, as some of the background aerosol information (e.g., biomass burning smoke) and pollution are advected into the interested study area. Another minor reason is for model evaluation purposes: to see if models with different resolutions can resolve the studied phenomena, as this is less studied and is of interest to check if models have the capability to represent them. The use of NAAPS and COAMPS provides insight into how aerosol-cloud interactions from in situ data are represented by coarse resolution models."

2) Synoptic conditions: I am concerned that by not providing some synoptic analysis, it is hard to understand why some of the results exist and why they differ from prior studies. Is it partially a sampling issue? If just 2 days have southerly flow but 10 do not for a given field campaign, the data for the 10 days should be more robust than for the 2 days? In Juliano et al. (2019, BAMS), this was addressed by selecting a sample of non-CTD days that was the same size as the CTD sample. If this was not done in the current study, it might be good to ensure that the datasets are consistent in this manner.

Response: Thank you for this comment. We have added some boundary layer flow maps to the Supplement (Fig. S13, 14, and 15) to provide some context. Additionally, we investigated how the aerosol characteristics changed when using the same number of northerly flow days as southerly flow days (selection of days and table of data shown below). The results present

generally the same trends as the larger selection of northerly flow days. Since the trend is consistent with the original table, we prefer to use the original in the manuscript. We think it is less biased by day-to-day variations and more representative of climatological proceedings.

| Campaign | Dates | Southerly Winds | Northerly Winds |
|---|---|---|---|
| E-PEACE | 07/08 – 08/18/2011 | 07/23, 24, 27, 28, 29 | 07/14, 15, 16 08/03, 04 |
| NiCE | 07/08 – 08/07/2013 | 07/16, 17, 18, 29 | 07/08, 09 08/01, 02 |
| BOAS | 07/02 – 07/24/2015 | 07/16, 17 | 07/06, 07 |
| FASE | 07/18 – 08/12/2016 | 07/29 | 08/04 |
| MACAWS | 06/21 – 07/12/2018 | 07/05, 12 | 06/21, 27 |
| CSM | 09/01 – 09/25/2020 | 09/01, 09, 10 | 09/15, 17, 21 |

| | $N_{a>10nm}$ | $N_{a10-100nm}$ | $N_{a0.1-1\mu m}$ | $N_{a>1\mu m}$ | $N_{a3}:N_{a10}$ | $N_d$ | Wind | Wind | $n_{Na}$ | $n_{Nd}$ | $n_{Wind}$ |
|---|---|---|---|---|---|---|---|---|---|---|---|
| | $(cm^{-3})$ | $(cm^{-3})$ | $(cm^{-3})$ | $(cm^{-3})$ | (-) | $(cm^{-3})$ | Speed (m s$^{-1}$) | Direction (°) | $(\times10^3)$ | $(\times10^3)$ | $(\times10^3)$ |
| E-PEACE | 861 / 336 | 501 / 262 | 338 / 88 | 0 / 0 | 1.09 / 1.11 | 252 / 59 | 3.38 / 8.19 | 177.61 / 327.39 | 20.3 / 44.3 | 17.1 / 22.0 | 37.4 / 66.7 |
| NiCE | 953 / 490 | 248 / 165 | 471 / 296 | 2.51 / 1.25 | 1.12 / 1.16 | 249 / 335 | 3.80 / 7.51 | 180.81 / 333.88 | 1.4 / 24.6 | 1.5 / 12.1 | 3.0 / 33.6 |
| BOAS | 750 / 383 | 553 / 213 | 204 / 174 | 0 / 0 | 1.20 / 1.18 | 143 / 100 | 5.49 / 3.61 | 166.97 / 321.11 | 5.8 / 20.1 | 3.9 / 0.1 | 11.8 / 20.0 |
| FASE | 836 / 679 | 423 / 267 | 326 / 241 | 0 / 0 | 1.29 / 1.13 | 203 / 320 | 2.35 / 7.01 | 144.03 / 320.57 | 1.0 / 10.0 | 0.3 / 6.3 | 1.3 / 16.2 |
| MACAWS | 722 / 1,011 | 560 / 821 | 154 / 186 | 0 / 1.25 | 1.25 / 1.24 | 189 / 215 | 7.75 / 9.38 | 162.15 / 316.20 | 10.3 / 17.1 | 6.6 / 6.5 | 16.9 / 23.7 |
| CSM | 5,558 / 660 | 5,081 / 536 | 515 / 176 | 1.00 / 0 | 1.30 / 1.25 | 334 / 142 | 6.10 / 3.70 | 193.93 / 330.28 | 4.8 / 6.8 | 1.8 / 0.7 | 6.9 / 13.3 |

We added the following text to the paper to mention this important caveat related to the choice of how to conduct the data analysis:

"This study's analysis focuses on maximizing the number of southerly and northerly cases available from the flight data rather than keeping a similar number of flights to represent southerly and northerly conditions. The rationale to include all available northerly flight days (which exceed southerly days; Table 1) is that their combined use is more representative of typical northerly conditions and less sensitive to inter-day variations. That being said, a random selection of northerly flight days was still used to compare to the more limited number of southerly flight days (not shown here), with the same generally conclusions reached as compared to using all northerly flight days."

Minor Comments

1) Figure 1: I realize that the flights for the different campaigns span different areas, but it is quite difficult to compare the regions that were sampled owing to the different axes used in each panel. I would suggest using a single set of axes for each plot for consistency. It would also be beneficial to know what percentage of each RF was i) over the open water as well as ii) in cloud. I say this because those are the samples that

are of interest to this study, and while the study does expand on the number of cases used in prior work, it is hard to tell exactly how much the dataset is expanded. Many of the flights appear to be largely over land, which would diminish the size of the relevant data.

Response: We have updated Figure 1 as shown below where each plot shares the same set of axes and provides the percentage of flights over the ocean and in cloud. There is no clear correspondence between the percentages reported below and the frequency of intercepting southerly days in these campaigns. Thanks for this suggestion as we agree it is nice to show these percentage values for context.

[Figure]

**Figure 1: Research flight paths for the six Twin Otter campaigns used in this study. The aircraft base at Marina, CA is denoted by a white diamond, and the IMPROVE station used in this study is indicated by a black star (Pt. Reyes National Seashore). The legends in each panel report on the percentage of flight time spent over the ocean and in cloud over the ocean.**

2) Line 205: This line makes references to thresholding data for when the aircraft was over the ocean, which further emphasizes my point made above about quantifying that for the reader.

Response: In addition to updating Fig. 1 according to the previous suggestion, we have also updated the text:

"The primary focus of the analysis is using data within the spatial domain listed in Sect. 2.1 only when the aircraft was over the ocean (Fig 1)."

3) Lines 232-233: How was this done exactly? I ask because assessing wind speed and direction from cloud motion works if the clouds follow the motion of the air, i.e., act as a tracer. In many cases, they do not act as a tracer of air motions, e.g., mountain wave clouds and the stratocumulus decks that are the focus of this study. So I think more details are needed here to provide confidence that the cloud motion extracted from the geostationary satellite data is in fact close to the wind speed/direction.

Response: Thank you for this comment. Since there were other sources with more robust data that we utilized to determine southerly flow, we have moved and updated the text:

"We also used Multi-Channel RGB data from the Geostationary Operational Environmental Satellite-WEST Full Disk Cloud Product (GOES-15) to investigate cloud motion on northerly and southerly flow days. The analysis utilized time resolutions of every three hours for E-PEACE, hourly for NiCE, BOAS, FASE, and MACAWS, and every half-hour for CSM.  We investigated all days within a campaign month, and not just days coinciding with a RF. For example, E-PEACE comprised flights from 9 July to 18 August 2011, and thus GOES data from 1 July through 31 August 2011 were investigated for that year. While not an exact tracer for air motion, we did observe that clouds tended to follow the prevalent air motion, particularly on southerly flow days."

4) Line 240: This location should be denoted on a map. Why was this location used?

Response: We have updated several places in the text to refer readers back to Fig. 1, where the location given in L240 is denoted on the map.

L123: "This study utilizes data from six airborne missions based out of Marina, CA (white diamond; Fig. 1)"

L240-242: "The National Oceanic and Atmospheric Administration (NOAA) Hybrid Single-Particle Lagrangian Integrated Trajectory (HYSPLIT; Stein et al., 2015; Rolph et al., 2017) model was also used to obtain back trajectories based on North American Mesoscale Forecast System (NAM) meteorological data (12 km resolution) ending at Marina, CA (36.67° N, 121.60° W; white diamond in Fig. 1) for 500, 900, 2,500, and 4,500 m AGL. Marina, CA was selected as the ending point for the back-trajectories as this was the takeoff/landing location for all six campaigns."

5) Line 260: I would suggest calling this the "meridional" or "north-south" wind and then highlight that north is positive.

Response: While we have opted to keep the variable as "northward wind speed" to stick with the NAAPS nomenclature, but we have added in this text for clarification:

"We investigated data for northward wind speed ($v_{wind}$, where northward (i.e., southerly) flow is indicated by positive values) and mass concentrations for ABF aerosols and sea salt (Fig. 2), along with smoke, dust, coarse aerosol, and fine aerosol (Fig. S2)."

6) Lines 263-265: Is there a sounding near the coast that you could use to show that this is a reasonable assumption?

Response: Thank you for this comment. We have added in the vertical profiles for median cabin temperature for each campaign, divided into northerly and southerly flow days, into the supplement (Fig. S3). The figure is provided below. This analysis is based on aircraft data. We have also added in this text:

"Vertical profiles of temperature for each campaign categorized by flow regime are provided in Fig. S3 using aircraft data over the ocean, to show the general structure of the lower troposphere in relation to the first five vertical levels of NAAPS."

We attempted to use data from the University of Wyoming's Department of Atmospheric Science at the Oakland station, however we opted to use the aircraft results as they are closer to our flight time (the soundings from Wyoming are only available at 0000 and 1200 UTC), and closer to our region where the reversals were being observed in this study.

[Figure]

**Figure S3: Median temperature vertical distributions for each campaign (northerly = blue; southerly = red). These data are based on aircraft measurements and are only over the ocean and screened to omit wildfire influence using criterion mentioned in Sect. 2.1.**

7) Lines 305-306: This is probably just my naiveness with these data, but why only every 3rd day?

Response: This is how the IMPROVE network works, which is to collect a 24 hr filter every third day. We have updated the text as follows just in case it helps clarify this better:

"Upon examination, it was decided to only use data for E-PEACE and BOAS because those campaign periods had more than a single point with valid data for southerly days (three and two, respectively); recall that IMPROVE data are only available every third day due to the sample collection procedure, so some southerly days would not necessarily have available IMPROVE data."

8) Lines 345-346: Could you plot the wind profiles to perhaps show that at the surface, the southerly flow is larger, but with height that flow weakens and leads to a mean decrease in wind speeds?

Response: Thank you for this comment. We have added this in text to the manuscript:

"All campaigns featured higher median wind speeds for northerly flow flights. However, when looking at the vertical wind profiles of each campaign for southerly and northerly flow days (Fig. S11), there were several instances where median wind speed at the surface for southerly flow days was greater than for northerly flow days."

We also added this figure to the supplement (Fig. S11):

[Figure]

**Figure S11: Median vertical wind speed profiles for each campaign (northerly = blue; southerly = red). These data are based on aircraft measurements and are only over the ocean and screened to omit wildfire influence using criterion mentioned in Sect. 2.1.**

9) Table 3 (and elsewhere): This may just be a personal thing but the use of "/" to separate values is sometimes confusing. I had to look back in a few instances to make sure that it was showing two different values and not intended to be a ratio. I might suggest being more explicit with the values and not use "/" even if the text becomes a bit longer.

Response: We appreciate this comment. We have seen slashes used in other publications (e.g., Crosbie at al., 2016), so we have chosen to keep Table 3 and other uses of the " / " in the tables, so no changes have been made to the manuscript.

10) Figure 2: I do not see a difference in the sign of the wind in the top left two panels. Both northerly and southerly flow have positive values? Is this just a mistake in the plotting?

Response: We have updated the color bar to try to show the difference more clearly in the northerly and southerly flows. NAAPS is not able to capture the southerly reversal due to its coarse grid size. See update in comment 14.

11) Line 361: The use of "Southerly–Northerly Days" is confusing. I am not sure if this is the transition day? A difference between southerly days and northerly days (I think that is what it is).

Response: We have updated this category to be labeled "Difference." See comment 14 for the updated figure.

Note this additional text in Sect. 3.1: " The $v_{wind}$ data are categorized into "Southerly Days," "Northerly Days," and "Difference" (i.e., southerly – northerly values)," as well as this sentence included in the caption for Fig. 2: "The right-most panel illustrates the difference between southerly and northerly flow days."

12) Lines 363-364: I am confused...it says above in the paper that by definition northerly is positive, so how could the northerly winds be less negative?

Response: We have revised this section as follows:

"Both southerly and northerly days had weaker $v_{wind}$ closer to the coast (up to 35° N) compared to farther offshore over the ocean (~ -3/-9 and -4/-6 m s$^{-1}$, respectively, for southerly/northerly flow). Slow, slightly northerly winds extended farther north to Marina and west to 123.5° W for southerly days, which is illustrated in red (differences exceeding ~3 m s$^{-1}$ between flow regimes) in the "Difference" panel. Northerly days also had an area of weaker $v_{wind}$ north of 43.5° N, which is emphasized in the "Difference" panel in blue (differences of -4 – -6 m s$^{-1}$)."

13) Lines 372-374: This sentence is very confusing; suggest rewording to better convey the meaning.

Response: We have revised this sentence as follows:

"A key conclusion from NAAPS is that the difference between southerly and northerly flow days matches expectations with southerly days having a greater tendency towards higher $v_{wind}$ compared to northerly days, but still not necessarily distinctly positive $v_{wind}$ values."

14) Figure 2 (again): Suggest using a colormap with white in the middle to make it easier to see positive vs negative values.

Response: Thank you for this comment. We have updated Fig. 2 as well as Fig. S4, which are provided below, respectively:

[Figure]

**Figure 2: Average northward wind speed ($v_{wind}$; m s$^{-1}$), total sea salt mass concentration (µg m$^{-3}$), and total ABF mass concentration (µg m$^{-3}$) of campaign months at 1800 UTC for 1$^{st}$ through 5$^{th}$ NAAPS levels (up to ~668 m above sea level) for southerly and northerly flow wind days. The right-most panel illustrates the difference between southerly and northerly**

**flow days. The airbase in Marina, CA is denoted by a white diamond, Pt. Reyes is indicated with a black star, and the black box indicates the region of focus in this study.**

[Figure]

**Figure S4: Average northward wind speed ($v_{wind}$: m s$^{-1}$) of campaign months (see Table 1) at 0000, 0600, and 1200 UTC for 1$^{st}$ through 5$^{th}$ NAAPS levels (up to ~668 m above sea level) for days identified as having southerly and northerly wind based on datasets described in Section 2.2. The color bar for the left two columns of panels can be interpreted as red values being for northerly flow and blue as southerly flow. The right-most panel illustrates the difference between southerly and northerly wind days. The airport in Marina, California is denoted by a white diamond, Pt. Reyes is indicated with a black star, and the black box outlines the region of focus in this study.**

15) Lines 385-387: Fair enough that in some cases, the fires were not right near the sampling site, but smoke can be advected for long distances, so how can you be certain that these campaigns were not affected by the more distant wildfires?

Response: We have added in this text for further context:

"These maps are mainly contextual to show the spatial distribution of fire sources and specific conclusions cannot be gleaned solely based on these regarding which campaigns had more or less wildfire influence overlapping with the flight tracks. This is especially the case because smoke can be advected from far distances away from the study region. The wildfire filter

described in Sect. 3.1 aims to filter out a large portion of smoke influence, at least at the regional level.”

16) Line 387: This is the first mention of “Marina”. Having taken part in EPEACE, I know what you are talking about, but the general audience may not. I would be more clear with where this is.

Response: We originally introduced Marina at the beginning of Sect. 2.1, and in that area we mention more about its proximity to the coastline:

"This study utilizes data from six airborne missions based out of Marina, CA (white diamond; Fig. 1) using the Naval Postgraduate School (NPS) Twin Otter aircraft. Marina is approximately 5 km away from the coastline.”

We also updated the text for the line in question to clarify this location:

“Past studies using airborne and surface-based data at Marina, CA (airbase indicated by a white diamond in Fig. 1 and 2) overlapping with the six campaigns in Table 1 revealed the following in terms of notable biomass burning influence around Marina and offshore areas”

17) Figure 4: Suggest adding a y axis label to say what is shown, not just the units.

Response: Thank you for this comment. We have updated Fig. 4 as follows:

[Figure]

**Figure 4: Box plots of IMPROVE data from the Pt. Reyes surface station. The southerly data for E-PEACE and BOAS (three and two points, respectively) are represented by the red boxes, and the northerly data (18 and seven, respectively) are represented by the blue boxes.**

18) Line 523: Define "reasonable"?

Response: We have updated the manuscript as follows:

"$SO_4^{2-}$, $NO_3^-$, OC, V, Ni, and EC are reasonable tracer species representative of either shipping and/or continental sources in the study region , as they have been utilized as tracers for these sources in previous studies (Wang et al., 2014; Maudlin et al., 2015; Wang et al., 2016; Dadashazar et al., 2019; Ma et al., 2019)."

19) Lines 561-562: Suggest adding a sentence or two at least speculating on why supermicron aerosol could change (i.e., put forth a hypothesis or two and then proceed with the analysis to confirm/deny).

Response: We have updated sections of the text as follows:

Hypothesis at outset of Sect. 3.2.3: "With all the complexities leading to sea salt emissions in the region (Schlosser et al., 2020), which is the predominant supermicron aerosol type in the study region's boundary layer, combined with the shifting wind directions and speeds leading up to and after a wind reversal (e.g., Juliano et al., 2019a), there was no underlying expectation for a change in levels during southerly flow events."

"The enhancement during southerly flow during at least CSM is presumed to be due to pervasive smoke during many of those RFs. However, the small median concentrations for each campaign make it hard to definitively determine if the lower concentrations during E-PEACE and BOAS were due to changes in flow regime or another factor."

"In general, the NAAPS results are consistent with aircraft and IMPROVE results in that in the study domain, there was not any pronounced difference in coarse aerosol characteristics during southerly flow. More research and data would be helpful though to put this conclusion on firmer ground."

20) Lines 599-606: This was something that we struggled with in Tim's papers was whether the aerosol were influencing the clouds or the meteorology was influencing the clouds? Any thoughts?

Response: Thank you for this comment. This is a great idea for future work. As this is not what we are attempting to answer in this manuscript, we have not made any changes to the manuscript.

21) Figure 5: It might be good to label the y axes and not just the top of the plots.

Response: Thank you for this comment. We have updated Fig. 5, as well as Fig. S12 and S16, as follows:

[Figure]

**Figure 5: Box plots of MODIS data within the study region during the periods overlapping with E-PEACE and BOAS. The southerly data for E-PEACE and BOAS (eight points each) are represented by the red boxes, and the northerly data (44 and 17 points, respectively) are represented by the blue boxes. The notches (and shading, which helps to more clearly indicate where the notches end) of the boxes assist in the determination of significance between multiple medians. If the notches overlap, the medians are not significantly different from one another.**

[Figure]

**Figure S12: Box plots of airborne data (Table 3) over the ocean in the study region during each campaign after removal of data when a flight's median $N_{a>10nm}$ exceeded 7,000 cm$^{-3}$. Northerly and southerly data are represented by blue and red boxes, respectively. The notches (and shading) of the boxes assist in the determination of significance between multiple medians. The p-values derived from Mann-Whitney U tests are reported in Table S4.**

[Figure]

**Figure S16: Box plots of cloud water speciated concentrations, along with Cl⁻:Na⁺ mass ratio and pH, within the study region during each campaign. The northerly data are represented by the blue boxes, and the southerly data are represented by the red boxes. The notches (and shading) of the boxes assist in the determination of significance between multiple medians. The p-values derived from Mann-Whitney U tests are reported in Table S5. No cloud water data are available for CSM.**

22) Lines 779-791: I feel like the modeling here distracts from the main theme. This also loops back to major comment number 1.

Response: Thank you for this comment. We have moved the model comparison to the supplement (with Fig. 10 now being Fig. S20). As a note, we have updated the COAMPS figure with model data that is similar to the altitude levels of the NAAPS results (660 and 668), and updated the text as well regarding the changes:

"Generally, NAAPS and COAMPS match in the general areas identified as having smoke and areas of high concentrations match one another. There is a small difference between the two models with respect to smoke (Fig. S20c and S20d), as COAMPS more closely matches the visible satellite imagery from Fig. 8b."

23) Lines 804-806: I think Juliano et al. (2019, JAS) did use all of these data sources, but maybe not in the same proportion as in the current study?

Response: We appreciate this comment. We have revised the text as follows:

"Our study is among the first to investigate aerosol and cloud droplet number concentrations through in situ aircraft data in addition to CW composition, and intercompare those results with satellite data, as well as models and surface station data. This builds upon previous studies, such as Juliano et al. (2019b), by utilizing similar data sources in greater proportions."

24) Figure S10: The wind speeds in the MACAWS case are quite strong, approaching 25 m/s based on Fig. S10. Any idea what was going on to cause such winds?

Response: Looking into these high wind speeds, they came from three particular RFs: 1, 2, and 14. When looking at the back-trajectories of these RFs, the highest two altitude levels investigated originated from over the open ocean, where wind speeds are typically higher than closer to shore. We did not feel this needed to be addressed in the manuscript, so no changes have been made.

[Figure]

[Figure]

We want to thank Dr. Lebo for the detailed examination of the manuscript and the SI file. We believe that both files have improved considerably thanks to these thoughtful comments. We also agree that paying more attention to the modeling aspects of this topic would be beneficial, and hope this study is one factor in many that leads others to pursue this.

---

## Referee Report (RR1)

**Second Review of "Differences in aerosol and cloud properties along the central California coast when winds change from northerly to southerly" – Zeider et al. (2024)**

The authors have done a thorough and commendable job revising the manuscript based on my comments/suggestions, as well as those of Dr. Witte. I have just one minor comment regarding the southerly vs. northerly flow in the model output compared with the observations.

If the authors have any questions, please do not hesitate to contact me.

Zachary J. Lebo

Minor Comment

1) I am still a bit concerned about the southerly vs. northerly winds shown in Fig. 2 from the model simulations compared with the differences shown in Table 3 from observations. The added note about the model not fully capturing the wind reversal is great, and it does help recognize that the model is perhaps a bit deficient here but at the very least the mechanics of the wind slowing are captured. One thing that comes to mind though about the difference between the observations shown in Table 3 and the model output shown in Fig. 2 is that perhaps there is a mismatch in the sampling of the vertical? The model output is averaged over the lowest 5 levels, does the complete lack of a wind reversal appear at all levels? I guess what I am after is trying to understand if the model predicts a change in the wind direction at some level or at some time, but when averaged spatially and temporally, the instances where the model does capture a reversal are overwhelmed by the instances where it does not.

---

## Referee Report (RR2)

Review 2, Zeider et al., EGUsphere/ACP

**SUMMARY**

I thank the authors for their work in responding to my comments. My major concerns have been adequately addressed. A few more minor points have arisen in the revised version that I ask the authors to respond to, at which point I believe the manuscript will be suitable for publication.

MKW

**MINOR COMMENTS**

*Difference figures*: We may have opened Pandora's box a little with the colormap switch in figures 2 and S4, but it strikes me that the difference plots you show in many of the right columns of these and other multi-panel figures (Figs. S2, S5-10) would also benefit from a colormap that shows a "neutral" shade for "no difference." Then you'd only be using the jet/rainbow colormap for scalar variables. I don't think this is a *requirement* but I strongly encourage you to give it a try and see if it improves interpretability.

*Boundary layer flow figures* (S13-15): These are a great addition. I have a suggestion based on my own experience making these kind of plots – instead of having the quivers show wind speed and direction, use line contours for wind speed and use the arrows to show *only* the wind direction. You can use the same plotting function to show wind direction, just normalize by vector magnitude so all the arrows are the same length. This is helpful because the southerly flow is often rather weak, so the tiny arrows force the reader to zoom to something crazy like 300% to be able to confirm that, yes, there is indeed a slight southerly component by the coast.

**TYPOGRAPHICAL COMMENTS**

L615-616: "there was no underlying expectation for a change in levels during southerly flow events" – unclear what you mean by "levels" here. I think you mean sea salt concentration, so maybe just change "levels" to "concentrations"

L878: "utilizing similar data sources in greater proportions" – the phrase "greater proportions" is ambiguous because "proportions" signifies a change in relative distribution. I think you're trying to say you use more cases than Juliano did, so I suggest rewording "in greater proportions" to "across a broader range of cases."

L879: "proven correct" is an overstatement. Suggest rewording to the following: "We find strong support for our first hypothesis that more final aerosol pollution…"

---

## Author Response (AR2)

We thank the reviewers for their helpful comments. We have provided our responses to comments below in blue.

REFEREE 1

The authors have done a thorough and commendable job revising the manuscript based on my comments/suggestions, as well as those of Dr. Witte. I have just one minor comment regarding the southerly vs. northerly flow in the model output compared with the observations.

If the authors have any questions, please do not hesitate to contact me.

Zachary J. Lebo

Response: We thank Dr. Lebo for their positive response to our changes.

**MINOR POINTS**

I am still a bit concerned about the southerly vs. northerly winds shown in Fig. 2 from the model simulations compared with the differences shown in Table 3 from observations. The added note about the model not fully capturing the wind reversal is great, and it does help recognize that the model is perhaps a bit deficient here but at the very least the mechanics of the wind slowing are captured. One thing that comes to mind though about the difference between the observations shown in Table 3 and the model output shown in Fig. 2 is that perhaps there is a mismatch in the sampling of the vertical? The model output is averaged over the lowest 5 levels, does the complete lack of a wind reversal appear at all levels? I guess what I am after is trying to understand if the model predicts a change in the wind direction at some level or at some time, but when averaged spatially and temporally, the instances where the model does capture a reversal are overwhelmed by the instances where it does not.

Response: This is a great comment. We went back and did some digging, and found that across the first five vertical levels, the magnitude and directionality of the northward wind was pretty consistent for any given day. Any changes observed in magnitude and directionality for a given day were across the four times we investigated: 0000, 0600, 1200, and 1800 UTC.

Looking at all of the southerly wind days, there were some instances of southerly wind across a full day at all five vertical levels, namely in MACAWS and CSM. However, as you theorized, these days were overwhelmed by instances where southerly wind was not captured by the model.

To address this, we have added in some text to the manuscript:

L396-401: "Generally, NAAPS was not able to fully capture southerly winds over the ocean and along the coast in that $v_{wind}$ was not clearly positive (i.e., not northward); however, when looking at southerly flow for individual campaigns, NAAPS was sometimes able to capture areas with

positive northward wind (i.e., southerly flow). When looking at the five vertical levels closest to the surface during periods when NAAPS was able to simulate positive northward winds, this feature was observed across all the levels, primarily along the coast near Marina, CA or south of 34° N at 1800 UTC, with lower wind speeds closer to the surface."

L854-856: "During cases when there was known southerly wind, NAAPS was only sometimes able to represent it, which is a topic encouraged for pursuit in future work."

We want to thank Dr. Lebo for bringing up this excellent question and believe that diving further into this topic would be a great aspect for a follow-up paper.

**REFEREE 2**

I thank the authors for their work in responding to my comments. My major concerns have been adequately addressed. A few more minor points have arisen in the revised version that I ask the authors to respond to, at which point I believe the manuscript will be suitable for publication.

MKW

Response: We thank Dr. Witte for the positive reply.

**MINOR COMMENTS**

*Difference figures*: We may have opened Pandora's box a little with the colormap switch in figures 2 and S4, but it strikes me that the difference plots you show in many of the right columns of these and other multi-panel figures (Figs. S2, S5-10) would also benefit from a colormap that shows a "neutral" shade for "no difference." Then you'd only be using the jet/rainbow colormap for scalar variables. I don't think this is a requirement but I strongly encourage you to give it a try and see if it improves interpretability.

Response: We have updated the following figures to use colormaps with a "neutral" shade for the "Difference" panels of the following figures:

[revised manuscript text omitted]

*Boundary layer flow figures* (S13-15): These are a great addition. I have a suggestion based on my own experience making these kind of plots – instead of having the quivers show wind speed and direction, use line contours for wind speed and use the arrows to show only the wind direction. You can use the same plotting function to show wind direction, just normalize by vector magnitude so all the arrows are the same length. This is helpful because the southerly flow is often rather weak, so the tiny arrows force the reader to zoom to something crazy like 300% to be able to confirm that, yes, there is indeed a slight southerly component by the coast.

Response: This is a great suggestion. We have adjusted Figures S13-15 accordingly:

[Figure]

**Figure S13: NAVGEM boundary layer flow patterns for (a) all southerly flow days and (b) all northerly flow days included in this study. These results are for 1800 UTC for all days of the campaign months in Table 1 as was done for Figs. S4-S10; the lowest level of the model was used representing the lowest ~50 m. The airport in Marina, California is denoted by a white diamond and Pt. Reyes is indicated with a white star.**

[Figure]

**Figure S14: NAVGEM boundary layer flow patterns for (left) southerly and (right) northerly flow days during (a-b) E-PEACE, (c-d) NiCE, and (e-f) BOAS. These results are for 1800 UTC for all days of these respective campaign months in Table 1; the lowest level of the model was used representing the lowest ~50 m. The airport in Marina, California is denoted by a white diamond and Pt. Reyes is indicated with a white star.**

[Figure]

**Figure S15: NAVGEM boundary layer flow patterns for (left) southerly and (right) northerly flow days during (a-b) FASE, (c-d) MACAWS, and (e-f) CSM. These results are for 1800 UTC for all days of these respective campaign months in Table 1; the lowest level of the model was used representing the lowest ~50 m. The airport in Marina, California is denoted by a white diamond and Pt. Reyes is indicated with a white star.**

TYPOGRAPHICAL COMMENTS

L615-616: "there was no underlying expectation for a change in levels during southerly flow events" – unclear what you mean by "levels" here. I think you mean sea salt concentration, so maybe just change "levels" to "concentrations"

Response: Change made.

L878: "utilizing similar data sources in greater proportions" – the phrase "greater proportions" is ambiguous because "proportions" signifies a change in relative distribution. I think you're trying to say you use more cases than Juliano did, so I suggest rewording "in greater proportions" to "across a broader range of cases."

Response: Change made.

L879: "proven correct" is an overstatement. Suggest rewording to the following: "We find strong support for our first hypothesis that more final aerosol pollution…"

Response: Change made.

In closing, we thank Dr. Witte for these additional comments, as they enhance and improve readability of the figures in the manuscript and supplement.